# Sample and Map from a Single Convex Potential: Generation using Conjugate Moment Measures

**Nina Vesseron**
CREST-ENSAE, IP Paris
nina.vesseron@ensae.fr

**Louis Béthune**
Apple
l_bethune@apple.com

**Marco Cuturi**
CREST-ENSAE, Apple
cuturi@apple.com

## Abstract

The canonical approach in generative modeling is to split model fitting into two blocks: define first *how* to sample noise (e.g. Gaussian) and choose next *what* to do with it (e.g. using a single map or flows). We explore in this work an alternative route that *ties* sampling and mapping. We find inspiration in *moment measures* [Cordero-Erausquin and Klartag, 2015], a result that states that for any measure $\rho$, there exists a unique convex potential $u$ such that $\rho = \nabla u \sharp e^{-u}$. While this does seem to tie effectively sampling (from log-concave distribution $e^{-u}$) and action (pushing particles through $\nabla u$), we observe on simple examples (e.g., Gaussians or 1D distributions) that this choice is ill-suited for practical tasks. We study an alternative factorization, where $\rho$ is factorized as $\nabla w^* \sharp e^{-w}$, where $w^*$ is the convex conjugate of a convex potential $w$. We call this approach *conjugate* moment measures, and show far more intuitive results on these examples. Because $\nabla w^*$ is the Monge map between the log-concave distribution $e^{-w}$ and $\rho$, we rely on optimal transport solvers to propose an algorithm to recover $w$ from samples of $\rho$, and parameterize $w$ as an input-convex neural network. We also address the common sampling scenario in which the density of $\rho$ is known only up to a normalizing constant, and propose an algorithm to learn $w$ in this setting.

## 1 Introduction

A decade after the introduction of GANs [Goodfellow et al., 2014] and VAEs [Kingma and Welling, 2014], the field of generative modeling has grown into one of the most important areas of research in machine learning. Both canonical approaches follow the template of learning a transformation that can map random codes to meaningful data. These transformations can be learned in supervised manner, as in the dimensionality reduction pipeline advocated in VAEs, or distributionally as advocated in the purely generative literature. In the latter, the variety of such transforms has gained remarkably in both complexity, using increasingly creative inductive biases, from Neural-ODEs[Chen et al., 2018, Grathwohl et al., 2018], diffusions [Song et al., 2020, Ho et al., 2020], optimal transport [Korotin et al., 2020] to flow-matching [Lipman et al., 2023, Tong et al., 2023, Pooladian et al., 2023].

**Sample first, move next.** All of these approaches are grounded, however, on choosing a standard Gaussian multivariate distribution to sample noise/codes. For both GANs and VAEs, that choice is usually made because of its simplicity, but also, in the case of diffusion models, because Gaussian distributions can be recovered quickly with suitable stochastic processes (Ornstein–Uhlenbeck). Other works have considered optimizing prior noise distributions [Tomczak and Welling, 2018, Lee et al., 2021, Liang et al., 2022] but still do so in a two step approach where mappings are estimated

39th Conference on Neural Information Processing Systems (NeurIPS 2025).

independently. We explore in this work a new generative paradigm where sampling and transforms are not treated separately, but seen as two facets of the same single convex potential.

**Tying sampling and action.** Our work looks to *moment measures* [Cordero-Erausquin and Klartag, 2015, Santambrogio, 2016] to find inspiration for a factorization that ties both sampling and action. Cordero-Erausquin and Klartag [2015] proved that for any sufficiently regular probability measure $\rho$, one can find an essentially-continuous, convex potential $u$ such that $\rho = \nabla u \sharp e^{-u}$: essentially, *any* probability distribution can be recovered by first sampling points from a log-concave distribution $e^{-u}$, and moving them with $\nabla u$ where, remarkably, the *same* single convex potential $u$ is used twice.

**Our Contributions.** We build on the work of Cordero-Erausquin and Klartag [2015] to propose an alternative path to build generative models, that we call *conjugate moment measures*.

- After recalling in detail the contribution of Cordero-Erausquin and Klartag [2015] in §2, we argue in §3.1 that the moment measure factorization may not be suitable for practical tasks. For instance, in the case of Gaussian distributions, we show that if $\rho$ is Gaussian with variance $\Sigma$, the corresponding log-concave distribution $e^{-u}$ is Gaussian with variance $\Sigma^{-1}$, amplifying beyond necessary any minor degeneracy in $\Sigma$. Similarly, for univariate $\rho$, we found that the spread of $e^{-u}$ is inversely proportional to that of $\rho$. While these results may not be surprising from a theoretical perspective, they call for a different strategy to estimate a tied sample/action factorization.

- We propose instead in §3.2 a new factorization that is conjugate to that of Cordero-Erausquin and Klartag [2015]: We show that any absolutely continuous probability distribution $\rho$ supported on a compact convex set can be written as $\rho = \nabla w^* \sharp e^{-w}$, where $w^*$ is the convex conjugate of $w$. Importantly, in our factorization the convex potential $w$ is used to map $\rho$ to $e^{-w}$, since one has equivalently that $\nabla w \sharp \rho = e^{-w}$. Note that evaluating $w^*$, the convex conjugate of $w$, only requires solving a convex optimization problem, since $w$ is assumed to be convex.

- We explore how to infer the potential $w$ from the conjugate factorization using either samples from $\rho$ or its associated energy function when the density of $\rho$ is only known up to a normalizing constant, leveraging optimal transport (OT) theory [Santambrogio, 2015]. Indeed, since the condition $\rho = \nabla w^* \sharp e^{-w}$ is equivalent to stating that $w^*$ is the Brenier potential linking $e^{-w}$ to $\rho$, we can parameterize $w$ as an input-convex neural network $w_\theta$ whose gradient is an OT map. We provide illustrations on simple generative examples.

## 2 Background

**Optimal Transport.** Let $\mathcal{P}(\mathbb{R}^d)$ denote the set of Borel probability measures on $\mathbb{R}^d$. For $\mu, \nu \in \mathcal{P}(\mathbb{R}^d)$, let $\Pi(\mu, \nu)$ denote the set of couplings between $\mu$ and $\nu$. We consider the primal OT problem,

$$\mathcal{W}_2^2(\mu, \nu) := \inf_{\pi \in \Pi(\mu, \nu)} \int_{\mathbb{R}^d \times \mathbb{R}^d} \tfrac{1}{2} \|x - y\|^2 \, \mathrm{d}\pi(x, y), \tag{1}$$

with squared-Euclidean cost. This problem admits a dual formulation:

$$\sup_{(f,g) \in \mathrm{L}^1(\mu) \times \mathrm{L}^1(\nu)} \int_{\mathbb{R}^d} f \mathrm{d}\mu + \int_{\mathbb{R}^d} g \mathrm{d}\nu, \text{ s.t. } f(x) + g(y) \leq \tfrac{1}{2}\|x - y\|^2, \forall (x, y) \in \mathbb{R}^d \times \mathbb{R}^d. \tag{2}$$

The functions $f$ and $g$ that maximize the expression in (2) are referred to as the Kantorovich potentials. Equation (1) can itself be seen as a relaxation of the Monge formulation of the OT problem,

$$\inf_{\substack{T:\mathbb{R}^d \to \mathbb{R}^d, \\ T \sharp \mu = \nu}} \int_{\mathbb{R}^d} \tfrac{1}{2} \|x - T(x)\|^2 \mu(\mathrm{d}x) \tag{3}$$

where $\sharp$ is the pushforward operator. An important simplification of (2) comes from the fact that solutions $f^\star, g^\star$ for equation (2) must be such that $\tfrac{1}{2}\|\cdot\|^2 - f^\star$ is convex. Using this parameterization, let $\mathcal{C}(\mathbb{R}^d)$ be the space of convex functions in $\mathbb{R}^d$. Brenier proved that in that case, a $T^\star$ solving problem (3) exists, and is the gradient of the convex function $\tfrac{1}{2}\|\cdot\|^2 - f^\star$, namely $\mathrm{Id} - \nabla f^\star$. Making a change of variables, $u = \tfrac{1}{2}\|\cdot\|^2 - f$ and using the machinery of cost-concavity [Santambrogio, 2015, §1.4], one can show that solving (2) is equivalent to computing the minimizer of:

$$\min_{\substack{u \in \mathcal{C}(\mathbb{R}^d) \\ u(0)=0}} \int_{\mathbb{R}^d} u(x) \, \mathrm{d}\mu(x) + \int_{\mathbb{R}^d} u^*(x) \, \mathrm{d}\nu(x) \tag{4}$$

where $u^*(y) := \max_{x \in \mathbb{R}^d} \langle x, y \rangle - u(x)$ is the convex-conjugate of $u$ and noting that we lift any ambiguity on $u$ by selecting the unique potential such that $u(0) = 0$. By denoting $\mathcal{B}(\mu, \nu)$ the Brenier potential solving problem (4) above, this results in $\nu = \nabla \mathcal{B}(\mu, \nu) \sharp \mu$.

**Neural OT solvers.** The goal of neural OT solvers is to estimate $\mathcal{B}(\mu, \nu)$ using samples drawn from the source $\mu$ and the target distribution $\nu$. Makkuva et al. [2020], Korotin et al. [2020] proposed methods that leverage input convex neural networks (ICNN), originally introduced by Amos et al. [2017], to parameterize the potential $u$ as an ICNN. A key challenge in these approaches is handling the Legendre transform $u^*$. To overcome this, a surrogate network can approximate $u^*$ with recent advances by Amos [2023] improving these implementations through amortized optimization.

**Moment Measures.** For a given convex function $u$, the moment measure of $u$ is defined as the pushforward measure $\nabla u \sharp \mathfrak{P}_u$ where $\mathfrak{P}_u$ is the log-concave (Gibbs) probability measure with density $\mathfrak{P}_u(x) := \frac{e^{-u(x)}}{\int_{\mathbb{R}^d} e^{-u(z)} \mathrm{d}z}$. The moment measure of $u$ is well defined when the quantity $\int_{\mathbb{R}^d} e^{-u(z)} \mathrm{d}z$ is positive and finite. While any measure $\rho$ supported on a compact set $K$ is guaranteed to be the moment measure of some convex potential $u$ [Santambrogio, 2016], Cordero-Erausquin and Klartag showed that $u$ is often discontinuous at the boundary $\partial K$. By studying the variational problem

$$\min_{u \in \mathcal{C}(\mathbb{R}^d)} \int u^* \mathrm{d}\rho - \ln \left( \int e^{-u} \right), \tag{5}$$

Cordero-Erausquin and Klartag [2015], proved that if $\rho$ is a measure with a finite first moment, a barycenter at 0 and is not supported on a hyperplane, then it is the moment measure of an essentially continuous potential (in the sense of Cordero-Erausquin and Klartag [2015]).

## 3 From Moment Measures to *Conjugate* Moment Measures

For a measure $\rho$, we call the convex function $u$ that satisfies $\nabla u \sharp \mathfrak{P}_u = \rho$ the moment *potential* of $\rho$. Additionally, we refer to the Gibbs distribution associated with such potential, $\mathfrak{P}_u$, as the Gibbs *factor* of $\rho$. We show in this section that moment measures are not well suited for generative modeling purposes, as they often produce a Gibbs factor that differs significantly from the target distribution $\rho$. This limitation is particularly evident in the case of Gaussian distributions, which motivates the introduction of an alternative representation: the *conjugate* moment measure.

### 3.1 Limitations of Moment Measures

By writing the KL divergence between $\rho$ and $\mathfrak{P}_{u^*}$ as $\mathrm{KL}(\rho \| \mathfrak{P}_{u^*}) = \int \ln(\rho) \mathrm{d}\rho + \int u^* \mathrm{d}\rho + \ln \left( \int e^{-u^*} \right)$ and applying this in the variational problem studied by Cordero-Erausquin and Klartag as referenced in (5), we obtain that the moment potentials of $\rho$ minimize the following problem:

$$\min_{u \in \mathcal{C}(\mathbb{R}^d)} \mathrm{KL}(\rho \| \mathfrak{P}_{u^*}) - \ln \left( \int e^{-u} \int e^{-u^*} \right). \tag{6}$$

Equation (6) reveals that the Gibbs distributions associated with the Legendre transform of $\rho$'s moment potentials, $\mathfrak{P}_{u^*}$, are the log-concave distributions closest to $\rho$, up to the regularization term $-\ln \left( \int e^{-u} \int e^{-u^*} \right)$. This quantity has been studied by Ball [1986] and Artstein-Avidan et al. [2004], who showed that for convex functions $u$ whose corresponding Gibbs factor $\mathfrak{P}_u$ has its barycenter at the origin, the term $-\ln \left( \int e^{-u} \int e^{-u^*} \right)$ attains its minimum value of $-d \ln(2\pi)$ when $\mathfrak{P}_u$ is Gaussian. A direct consequence of (6) is that the Gibbs factor of $\rho$ can differ significantly from $\rho$, particularly when $\rho$ is Gaussian, as formalized in the following proposition:

**Proposition 1.** *Let* $\rho = \mathcal{N}(0_{\mathbb{R}^d}, \Sigma)$. *If* $\Sigma$ *is non degenerate, the moment potentials of* $\rho$ *are* $u_m(x) = \frac{1}{2}(x - m)^T \Sigma (x - m)$, *with* $m \in \mathbb{R}^d$. *The associated Gibbs factor of* $\rho$ *is* $\mathfrak{P}_{u_m} = \mathcal{N}(m, \Sigma^{-1})$.

*Proof.* The optimization problem in equation (6) is translation invariant: replacing $u$ with $x \mapsto u(x + a)$ for any $a \in \mathbb{R}^d$ leaves the objective unchanged. We can thus restrict to convex functions $u$ such that $\mathfrak{P}_u$ has barycenter at zero. In this setting, choosing $u(x) = \frac{1}{2} x^\top \Sigma x$ yields $\mathfrak{P}_u = \mathcal{N}(0, \Sigma^{-1})$ and $\mathfrak{P}_{u^*} = \mathcal{N}(0, \Sigma)$, which minimize the first term in (6), while the second term reaches its minimum value of $-d \ln(2\pi)$ [Artstein-Avidan et al., 2004]. Hence, this choice minimizes the full objective, and translations of $u$ span the set of moment potentials of $\rho$. $\square$

Intuitively this results can be interpreted as follows. When a peaked distribution $\rho$ is centered around the origin (e.g. $\Sigma \approx 0$), its corresponding moment potential is such that the *image* set $\nabla u$ on the support of $\mathfrak{P}_u$ is necessarily tightly concentrated around 0. This has the implication that $u$ is a *slowly* (almost constant) varying potential on the entire support of $\mathfrak{P}_u$. As a result, one has the (perhaps) counter-intuitive result that the more peaky $\rho$, the more spread-out $\mathfrak{P}_u$ must be. From this simple observation–validated experimentally in Figure 2–we draw the intuition that a change is needed to reverse this relationship, while still retaining the interest of a measure factorization result.

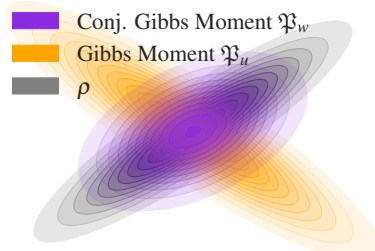

Figure 1: **Gibbs factor** and **conjugate Gibbs factor** of $\rho = \mathcal{N}\left(0, \begin{bmatrix} 2 & 1.8 \\ 1.8 & 2 \end{bmatrix}\right)$.

## 3.2 The *Conjugate* Moment Measure factorization

Our main result, which establishes the conjugate moment measure factorization, is stated below:

**Theorem 1.** *Let $\rho \in \mathcal{P}(\mathbb{R}^d)$ be an absolutely continuous probability measure supported on a compact, convex set. Then, there exists a convex function $w$ such that $\rho = \nabla w^* \sharp \mathfrak{P}_w$.*

Accordingly, we now refer to $\rho$ as the *conjugate moment measure* of $w$, $w$ as the *conjugate* moment potential of $\rho$, and $\mathfrak{P}_w$ as the *conjugate* Gibbs factor of $\rho$. Our proof strategy differs significantly from that used in Cordero-Erausquin and Klartag [2015] and Santambrogio [2016] since we use Schauder's fixed point theorem to show that the following map admits a fixed point:

$$G_\rho : \mathcal{L}(\mathbb{R}^d) \to \mathcal{L}(\mathbb{R}^d), \quad G_\rho(w) := \mathcal{B}(\rho, \mathfrak{P}_w) \tag{7}$$

where $\mathcal{L}(\mathbb{R}^d)$ is the set of functions mapping $\mathbb{R}^d$ to $\mathbb{R} \cup \{+\infty\}$, and $G_\rho$ assigns to a potential $w$ the Brenier potential transporting $\rho$ to $\mathfrak{P}_w$. Fixed points of $G_\rho$ correspond precisely to the conjugate moment potentials of $\rho$, and their existence guarantees a solution to the moment measure factorization. A sketch of the proof is provided in Appendix A.1, with full details in Appendix A.2. When $\rho$ is Gaussian, the conjugate moment measure factorization admits an explicit solution, as given in Proposition 2. Recall that we saw in §3.1 that Gibbs factors of $\mathcal{N}(0, \Sigma)$ were multivariate normal distributions with covariance matrix $\Sigma^{-1}$.

**Proposition 2.** *When $\Sigma$ is non-degenerate, $\rho = \mathcal{N}(m, \Sigma)$ is the conjugate moment measure of $w(x) = \frac{1}{2}(x - r)^T \Sigma^{-1/3}(x - r)$, whose Gibbs distribution is $\mathfrak{P}_w = \mathcal{N}(r, \Sigma^{1/3})$ where $r = (I_d + \Sigma^{1/3})^{-1}m$. The function $w$ is the unique conjugate moment potential of $\rho$ whose Gibbs distribution remains Gaussian.*

The proof is provided in Appendix A.4 and relies on the fact that the OT map between two Gaussians is known in closed form [Peyré et al., 2019]. While this potential $w$ may not be the only conjugate moment potential, the Gibbs distribution associated with it, $\mathcal{N}(r, \Sigma^{1/3})$, appears to be better suited to the target distribution $\rho$. This is particularly evident in the example shown in Figure 1. Additionally, it is worth noting that for the Gaussian case, our factorization still works when $\rho$ is not centered, whereas the approach of Cordero-Erausquin and Klartag [2015] *requires* $\rho$ to be mean 0.

## 3.3 Monge-Ampère Equation for Conjugate Moment Measures

From the moment measure factorization, $\rho = \nabla u \sharp \mathfrak{P}_u$, which holds at the level of probability distributions, one can derive the corresponding equality between probability density functions and obtain the following Monge–Ampère equation [Cordero-Erausquin and Klartag, 2015]:

$$\rho(x) = e^{-\mathcal{E}_u(x)} \quad \text{with} \quad \mathcal{E}_u(x) = u(\nabla u^*(x)) - \ln(\det H_{u^*}(x)) + \ln(C_u), \tag{8}$$

where $u^*$ is the convex conjugate of $u$, $H_{u^*}(x)$ denotes the Hessian of $u^*$ at $x$, and $C_u = \int e^{-u}$ is the normalizing constant ensuring that $\mathfrak{P}_u = e^{-u}/C_u$ is a probability distribution. Similarly, Theorem 1 guarantees the existence of a convex function $w$ that satisfies the following Monge-Ampère equation for any absolutely continuous probability measure $\rho$ supported on a compact, convex set:

$$\rho(x) = e^{-\mathcal{E}_w(x)} \quad \text{where} \quad \mathcal{E}_w(x) = w(\nabla w(x)) - \ln(\det H_w(x)) + \ln(C_w), \tag{9}$$

with $H_w(x)$ denoting the Hessian of $w$ at $x$ and $C = \int e^{-w}$ being the normalizing constant of $\mathfrak{P}_w$. The potentials $w$ that solve (9) are precisely the conjugate moment potentials of $\rho$.

# 4 Estimating Conjugate Moment Potentials in Practice

We first explain how to sample from $\rho$ when one of its conjugate moment potential is known, using the Langevin Monte Carlo (LMC) algorithm and a conjugate solver. We then describe a method to estimate a conjugate moment factorization of $\rho$ using i.i.d samples $(x_1, \ldots, x_n) \sim \rho$. In this approach, the conjugate moment potential of $\rho$ is parameterized using an input convex neural network (ICNN) $w_\theta$ following the architecture proposed in Vesseron and Cuturi [2024]. The conjugate moment potential $w_\theta$ is then estimated using an algorithm inspired by the fixed-point method associated to Theorem 1. Finally, we address the case where the density of $\rho$ is known up to a normalizing constant, a common scenario in sampling; we use an ICNN to parameterize the potential and estimate it via regression using the Monge–Ampère equation (9).

## 4.1 Sampling from $\rho$ using its Conjugate Moment Factorization

In this paragraph, we suppose that we know a conjugate moment potential $w$ of $\rho$, i.e. $\nabla w^* \sharp \mathfrak{P}_w = \rho$. Knowing $w$, drawing samples from $\rho$ can be done by first sampling $x \sim \mathfrak{P}_w$ and then applying $\nabla w^*$ to those points as $\nabla w^*(x) \sim \nabla w^* \sharp \mathfrak{P}_w = \rho$. The LMC algorithm is a widely used method for generating samples from a smooth, log-concave density like $\mathfrak{P}_w$ [Roberts and Tweedie, 1996, Cheng and Bartlett, 2018, Dalalyan and Karagulyan, 2019]. Starting from an initial point $x^{(0)}$, the LMC algorithm iterates according to the following update rule:

$$x^{(k+1)} = x^{(k)} - \tau \nabla w(x^{(k)}) + \sqrt{2\tau} z^{(k)}, \quad z^{(k)} \sim \mathcal{N}(0, I_d),$$

where $\tau$ is the step size. As for the gradient of the convex conjugate $\nabla w^*$, it can be efficiently estimated from $w$. By applying Danskin's envelope theorem [1966], it follows that $\nabla w^*(y)$ is the solution to the following concave maximization problem:$\nabla w^*(y) = \arg \sup_x \langle x, y \rangle - w(x)$. This optimization problem can be solved using algorithms such as gradient ascent, (L)BFGS [Liu and Nocedal, 1989], or ADAM [Kingma and Ba, 2014]. Thus, having access to a conjugate moment potential $w$ of $\rho$ enables to efficiently draw samples from it. Note that the gradient steps of the conjugate solver can be interpreted as a denoising procedure applied to samples drawn from $\mathfrak{P}_w$.

## 4.2 Learning the Conjugate Moment Factorization from Samples: CMFGen

We consider the case where $\rho \in \mathcal{P}(\mathbb{R}^d)$ can only be accessed through samples, as in an empirical distribution approximation $\rho_n := \frac{1}{n} \sum_{i=1}^n \delta_{x_i}$. The fixed points of the map $G_\rho$, as defined in (7), correspond exactly to the conjugate moment potentials of $\rho$. This observation motivates the following fixed-point iteration scheme to compute a conjugate moment potential of $\rho$:

$$w_0 := \tfrac{1}{2} \| \cdot \|^2; \quad \forall t \geq 1, \quad w_{t+1} := G_\rho(w_t), \tag{10}$$

Starting from $w_0 = \frac{1}{2} \| \cdot \|^2$, which corresponds to $\mathfrak{P}_{w_0} = \mathcal{N}(0, I_d)$, we iteratively compute the Brenier potential $w_{t+1}$ between the distribution $\rho$ and $\mathfrak{P}_{w_t}$ at iteration $t + 1$. During the next iteration $t + 2$, the updated distribution $\mathfrak{P}_{w_{t+1}}$ becomes the target distribution to compute the next Brenier potential starting from the source $\rho$. This process is repeated until the algorithm converges.

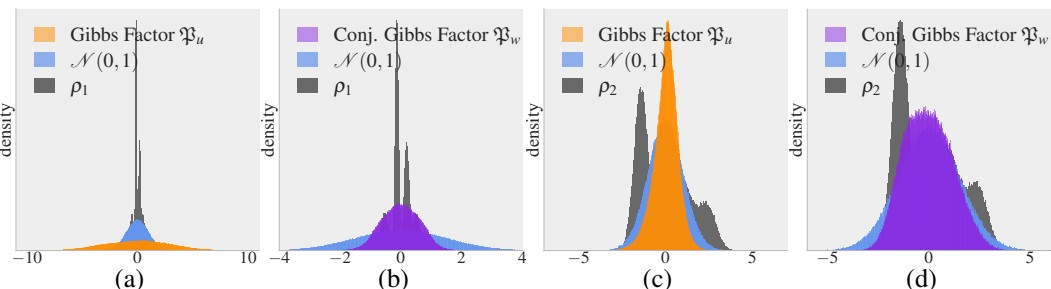

Figure 2: Comparison between the **Gibbs factor** $\mathfrak{P}_u$ and the **conjugate Gibbs factor** $\mathfrak{P}_w$ for two mixtures of 1D Gaussian distributions, $\rho_1$ and $\rho_2$. The density plots overlay the (conjugate) Gibbs factor with $\rho$ and a standard Gaussian $\mathcal{N}(0, 1)$ for reference. Gibbs factors spread inversely to $\rho$ (**(a), (c)**) while conjugate Gibbs factors show more suitable alignment (**(b), (d)**).

**Univariate distributions**  In the 1D case, the OT map between probability density functions $\mu$ and $\nu$ has a closed-form expression given by Peyré et al. [2019]:

$$\nabla \mathcal{B}(\mu, \nu) = C_\nu^{-1} \circ C_\mu$$

where $C_\mu : \mathbb{R} \to [0, 1]$ is the cumulative distribution function associated to $\mu$, defined as: $C_\mu(x) := \int_{-\infty}^x \mathrm{d}\mu$. The quantile function $C_\mu^{-1} : [0, 1] \to \mathbb{R} \cup \{-\infty\}$ is the pseudoinverse $C_\mu^{-1}(r) := \min\{x \in \mathbb{R} \cup \{-\infty\} : C_\mu(x) \geq r\}$. For a given distribution $\rho$ and an initial distribution $\mathfrak{P}_{w_0}$, the analytical form of the Brenier map enables the execution of the algorithm in (10). To compare the conjugate moment measure factorization with the moment measure factorization in 1D, we also propose a similar 1D algorithm for estimating a moment measure potential of a distribution $\rho$, described in Appendix B.1, as no method currently exists for solving the moment measure factorization. The code for both algorithms is also provided in Appendix B.1.

**Higher dimensional case**  In higher dimensions, where the OT map is not available in closed form, we estimate it using ICNNs with the architecture used in Vesseron and Cuturi [2024] and the neural OT solver proposed in Amos [2023]. We found that performing a single optimization step to estimate the OT map at each iteration was sufficient. As a result, our final methodology relies on a single ICNN, $w_\theta$, which serves two purposes at each iteration: generating samples from $\mathfrak{P}_{w_\theta}$ and being optimized by a single step of the neural OT solver to approximate the OT map between $\rho$ and $\mathfrak{P}_{w_\theta}$. Algorithm 1 details the steps of the procedure, where $\tilde{x}(y_i)$ is the estimation of $\nabla w_\theta^*(y_i)$, computed as described in Amos [2023]. The LMC sampling step in step 4 of Algorithm 1 requires selecting the step size hyperparameter $\tau$, which depends on $w_\theta$.

---

**Algorithm 1** CMFGen algorithm

1: Initialize $w_\theta$ such that $w_\theta \approx \frac{1}{2}\|\cdot\|^2$
2: **while** not converged **do**
3:     Draw $n$ i.i.d samples $x_i \sim \rho$
4:     Draw $y_1, \dots, y_n \sim \mathfrak{P}_{w_\theta}$ using LMC
5:     $\mathcal{L}_\theta \leftarrow \frac{1}{n}\sum_{i=1}^n w_\theta(x_i) - \frac{1}{n}\sum_{i=1}^n w_\theta(\tilde{x}(y_i))$
6:     Update $w_\theta$ with $\nabla \mathcal{L}_\theta$
7: **end while**

---

Since $w_\theta$ evolves during the algorithm, the step size must be dynamically adjusted. Following Proposition 1 in Dalalyan [2016], we set $\tau = \frac{1}{M_\theta}$, where $M_\theta$ is the largest eigenvalue of the Hessian of $w_\theta$, estimated from the current minibatch $\{x_1, \dots, x_n\}$. To accelerate convergence, the LMC algorithm is initialized using particles from the previous iteration.

### 4.3  Learning the Conjugate Moment Factorization from an Energy: CMFMA

We now consider the case where the density of $\rho$ is known only up to a normalizing constant, which is the typical setting in sampling-based frameworks. Specifically, we assume access to an energy function $\mathcal{E}$ such that $\rho \propto e^{-\mathcal{E}}$. To learn a conjugate moment potential of $\rho$, we propose to leverage the Monge–Ampère formulation defined in (9), and parameterize the potential using an ICNN $w_\theta$ trained via regression on the following objective:

$$\mathbb{E}_{x \sim \mathbb{P}}\left[\|\mathcal{E}(x) - w_\theta(\nabla w_\theta(x)) + \ln(\det H_{w_\theta}(x))\|^2\right].$$

In this objective, $\mathbb{P}$ can be taken as the uniform distribution over the sampling space when the dimension is small and the domain is bounded. In higher-dimensional settings, $\mathbb{P}$ can instead be chosen as $\rho$, with samples obtained via the Langevin Monte Carlo (LMC) algorithm. At the end of training, the learned potential $w_\theta$ satisfies $w_\theta(\nabla w_\theta(x)) + \ln(\det H_{w_\theta}(x)) \approx \mathcal{E}(x)$ and the pushforward $\nabla w_\theta^* \sharp \mathfrak{P}_{w_\theta}$ provides a close approximation of the target $\rho$. §4.1 details how to sample from $\nabla w_\theta^* \sharp \mathfrak{P}_{w_\theta}$ using only the learned potential $w_\theta$. Note that, in contrast, learning the moment potential $u$ via regression seems particularly challenging due to the presence of the Legendre transform $u^*$ in multiple terms of the Monge–Ampère equation (8).

## 5  Experiments

We begin with preliminary experiments using the CMFGen and CMFMA algorithms introduced in Section 4.2 and 4.3, starting with univariate distributions $\rho$ for which the Monge map is available in closed form. We then estimate the conjugate moment potential for several 2D distributions, either from samples (using CMFGen) or from an energy function (using CMFMA). Finally, we demonstrate the applicability of CMFGen to higher-dimensional datasets such as MNIST [LeCun et al., 2010] and

the Cartoon dataset from Royer et al. [2018]. At the moment, CMFGen is not comparable to state-of-the-art generative models such as flow matching [Lipman et al., 2023, Albergo and Vanden-Eijnden, 2023], and we believe this is primarily due to CMFGen's reliance on ICNNs, which, while useful to parameterize convex functions, are known to be challenging to train [Korotin et al., 2021]. For this reason, we restrict our comparisons to a generative ICNN trained to transport the Gaussian $\mathcal{N}(0, I)$ to the distribution of data $\rho$. We use the Sinkhorn divergence as a metric for the 2D experiments, and provide images generated by both our approach CMFGen and generative ICNNs.

## 5.1 Univariate distributions

We compute the *conjugate* moment potential for two univariate Gaussian mixtures, $\rho_1$ and $\rho_2$, shown in Figure 2, using both CMFGen and CMFMA. CMFGen uses i.i.d. samples from the mixture, while CMFMA leverages the log-density. The distribution $\rho_1$ is sharply concentrated around zero, whereas $\rho_2$ exhibits heavier tails. For comparison, we also compute the standard moment potentials using the fixed-point method described in Appendix B.1. The pushforward densities $\nabla w^* \sharp \mathfrak{P}_w$ and $\nabla u \sharp \mathfrak{P}_u$ (Figures 11 and 8) closely match the target distributions, confirming that the three algorithms successfully recover the (conjugate) moment potentials for $\rho_1$ and $\rho_2$. The recovered conjugate potentials are shown in Figures 9 and 10. As illustrated in Figure 2 (a), the Gibbs factor associated with $\rho_1$ has heavier tails than the standard Gaussian, consistent with the theoretical insight that concentrated distributions yield broader Gibbs measures. Conversely, the broader $\rho_2$ induces a more concentrated Gibbs factor (panel (c)). In contrast, the *conjugate* Gibbs factors (panels (b) and (d)) more closely match the target distributions.

## 5.2 2D Experiments

We consider several 2D distributions defined either through samples—(a) Circles, (b) S-curve, (c) Checkerboard, (d) Scaled-Rotated S-curve, and (e) Diag-Checkerboard (Figure 4)—or through known energy functions $\mathcal{E}_1$, $\mathcal{E}_2$, and $\mathcal{E}_3$ (Figures 3 and 12). Our first step is to estimate a conjugate moment potential $w_\theta$ for these distributions using either the CMFGen or the CMFMA algorithm. Following this, we generate new samples based on the learned conjugate potential $w_\theta$ using the methodology detailed in §4.1. The potential $w_\theta$ is implemented as an ICNN with five hidden layers of size 128 and quadratic input connections, based on the architecture of Vesseron and Cuturi [2024]. Detailed hyperparameters for both 2D and high-dimensional experiments are provided in Appendix E.

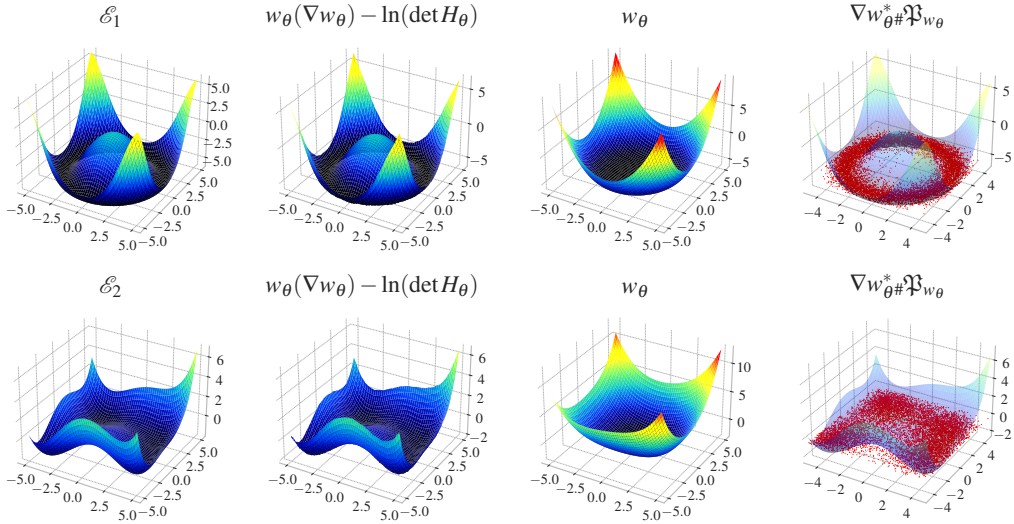

Figure 3: **Learning the conjugate moment potential from an energy**. $\mathcal{E}_1$ and $\mathcal{E}_2$ are learned by regression with CMFMA. The second column shows the learned energy; the third displays the corresponding conjugate moment potential; the fourth shows samples (in red) drawn from $\nabla w_\theta^* \sharp \mathfrak{P}_{w_\theta}$.

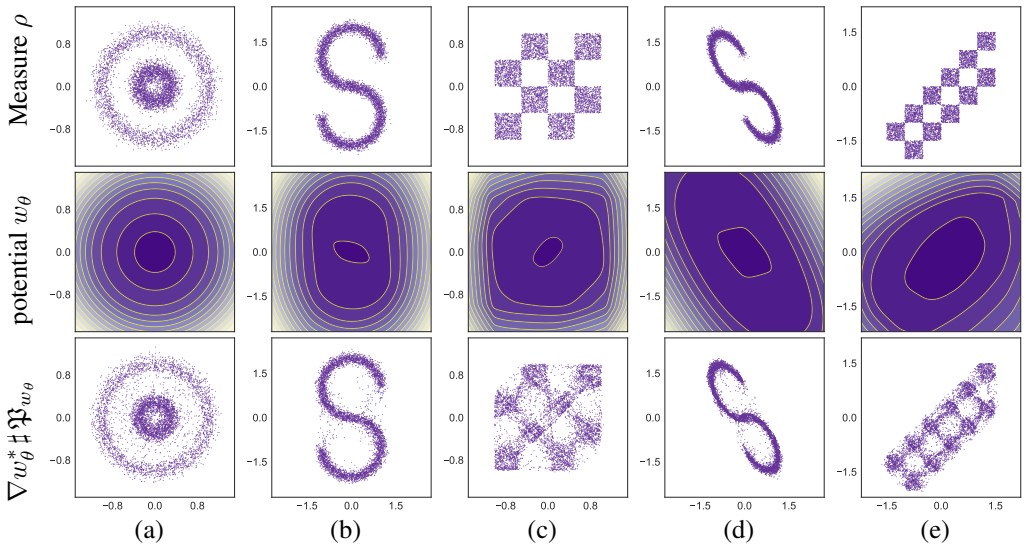

Figure 4: Samples from $\rho$ (**top**), level sets of $w_\theta$ (**middle**) and samples from $\nabla w_\theta^* \sharp e^{-w_\theta}$ (**bottom**).

**CMFMA.** The energy functions $\mathcal{E}_1$, $\mathcal{E}_2$, and $\mathcal{E}_3$ are standard 2D benchmarks for evaluating optimization algorithms; their analytical forms are provided in Appendix C.2. For all three, the ICNNs $w_\theta$ successfully learn the corresponding energy landscape and permit to draw new samples from the target distributions, as demonstrated in Figure 3 and 12.

**CMFGen.** Our method accurately estimates the conjugate moment potentials, as the associated measures $\nabla w_\theta^* \sharp \mathfrak{P}_{w_\theta}$ closely align with the target distributions in Figure 4. The second row further illustrates that the conjugate Gibbs factor follows the shape of the distribution $\rho$ in each case. For comparison, we train an ICNN to map a Gaussian directly to the target distribution using the solver of Amos [2023]. The boxplots in Figure 13, which show the Sinkhorn divergence [Ramdas et al., 2017] between generated and target data, demonstrate that CMFGen consistently produces samples of equal or superior quality. Note that CMFGen introduces no additional hyperparameters compared to the generative ICNN, aside from the number of LMC steps used when sampling from $\mathfrak{P}_{w_\theta}$.

### 5.3 High-dimensional experiments.

**Image generation.** We evaluate CMFGen on MNIST [LeCun et al., 2010] and the Cartoon dataset [Royer et al., 2018]. As illustrated in Figures 5, and 6, (see also 14 for additional generated cartoons), CMFGen successfully generates visually coherent images of digits and cartoon faces. Interestingly, the noise sampled from the log-concave distribution $\mathfrak{P}_{w_\theta}$ already exhibits features of the generated distribution: digits and faces emerge directly in the noise. To the best of our

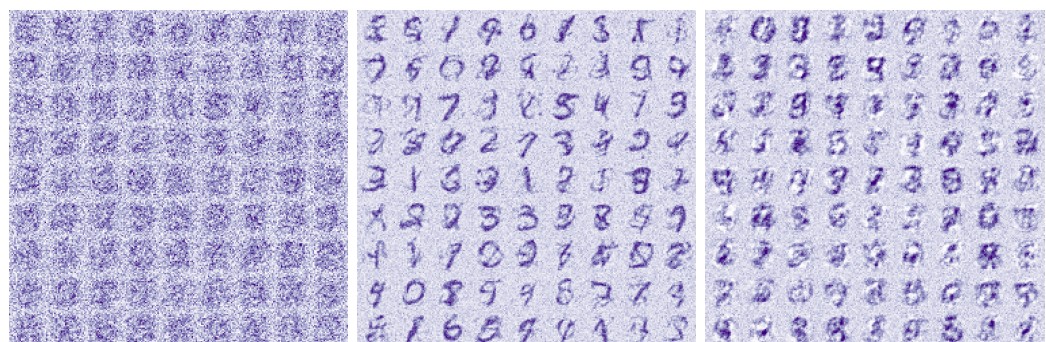

Figure 5: **MNIST Generation using CMFGen.** Samples from the Gibbs noise distribution $\mathfrak{P}_{w_\theta}$ (**left**); digits generated from $\nabla w_\theta^* \sharp \mathfrak{P}_{w_\theta}$ (**middle**); and digits generated by an ICNN trained to directly transport Gaussian noise to MNIST (**right**).

Gibbs $\mathfrak{P}_{w_\theta}$ obtained from Langevin dynamics $\quad\approx\quad$ Gibbs $\mathfrak{P}_{w_\theta}$ obtained from $\nabla w_\theta \sharp \rho$

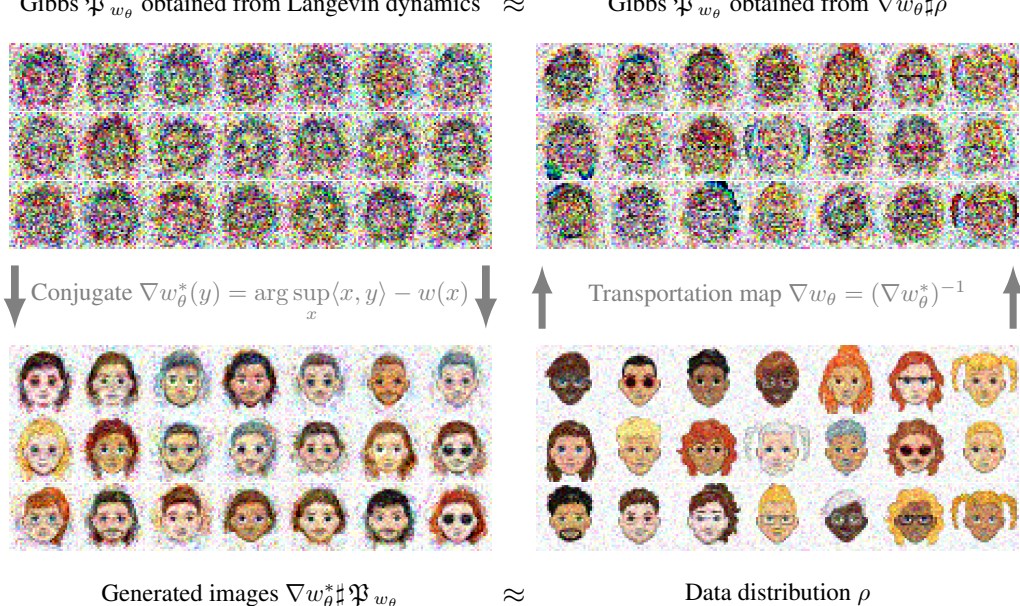

Conjugate $\nabla w_\theta^*(y) = \arg\sup_x \langle x, y \rangle - w(x)$ $\qquad$ Transportation map $\nabla w_\theta = (\nabla w_\theta^*)^{-1}$

Generated images $\nabla w_\theta^* \sharp \mathfrak{P}_{w_\theta}$ $\qquad\approx\qquad$ Data distribution $\rho$

Figure 6: **Cartoon Generation using CMFGen.** The conjugate potential $w_\theta$ is parameterized as an ICNN following Vesseron and Cuturi [2024], with five hidden layers of size 512 and four quadratic input connections, each with two additional layers of size 512. After training with CMFGen, the generative map $\nabla w_\theta$ transforms structured data $\rho$ into the log-concave distribution $\mathfrak{P}_{w_\theta}$. To sample from $\rho$, we first draw a sample from $\mathfrak{P}_{w_\theta}$ using the LMC algorithm, and then apply a conjugate solver to iteratively invert the map $\nabla w_\theta$. The strict convexity of $w_\theta$ ensures both **(i)** invertibility of $\nabla w_\theta$, and **(ii)** correctness of Langevin dynamics for sampling from $\mathfrak{P}_{w_\theta}$.

knowledge, this is the first instance where the MNIST distribution has been successfully generated using an ICNN. For comparison, Figures 5 and 15 show samples generated by ICNNs trained to transport a Gaussian distribution to the MNIST and Cartoon data. For both datasets, CMFGen generates samples of higher quality.

**Image reconstruction.** Similar to normalizing flows [Rezende and Mohamed, 2015], we have access to the (unnormalized) probability density of the distribution $\nabla w_\theta^* \sharp \mathfrak{P}_{w_\theta}$ generated by CM-FGen and CMFMA: that density is proportional to $e^{-\mathcal{E}_{w_\theta}(x)}$ where $\quad \mathcal{E}_{w_\theta}(x) = w_\theta(\nabla w_\theta(x)) - \ln(\det H_{w_\theta}(x))$ (see §3.3). After training $w_\theta$, this enables downstream tasks such as image inpainting. To evaluate this, we mask half of the pixels in test samples from MNIST and the Cartoon

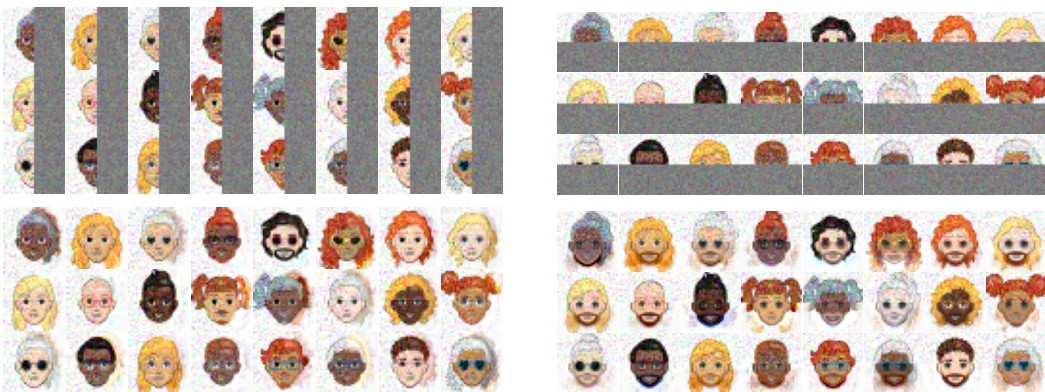

Figure 7: **Image inpainting task on Cartoon.** The learned $w_\theta$ trained on the Cartoon dataset is used for a post-processing task to recover the masked pixels. Gradient ascent is performed on the masked pixels to maximize the log-probability of the full image. Top: Masked images; Bottom: Reconstructed images.

dataset, and perform gradient ascent on masked pixels to maximize the log-probability of the full image. As seen in Figures 7, 16 and 17, our method effectively reconstructs missing regions.

# 6    Conclusion

We borrowed inspiration from Cordero-Erausquin and Klartag [2015] to define a new representation for probability measures. For a given $\rho$, we prove the existence of a convex potential $w$ such that $\nabla w^* \sharp \mathfrak{P}_w = \rho$. We show that this representation is better suited to generative modeling, because the Gibbs factor $\mathfrak{P}_w$ follows more closely the original measure $\rho$, in contrast to Cordero-Erausquin and Klartag's approach, $\nabla u \sharp \mathfrak{P}_u = \rho$, which results in a Gibbs factor $\mathfrak{P}_u$ whose spread is inversely proportional to that of $\rho$. Our conjugate measure factorization uses $w$ to sample noises (using LMC on a log-concave distribution) and transforms these codes in a final step, using $\nabla w^*$. We propose to parameterize the conjugate potential $w$ as an ICNN $w_\theta$, and estimate it using the OT toolbox in two settings: when the target distribution is accessible via samples, and when it is known up to a normalizing constant. We validate both approaches on generative modeling tasks. In the future, we wish to explore the suitability of replacing $\mathcal{N}(0, I)$ with our pre-trained Gibbs factor $\mathfrak{P}_w$ in generative modeling pipelines, and retrain maps on top of it. Interestingly, one can draw parallels between CMFGen and flow matching in the sense that CMFGen does a simple noisy gradient flow to generate codes (the LMC algorithm), concluded by a one step generation step $\nabla w_\theta^*$, which can be compared to the iterated application of a time-varying (non-conservative) velocity field to generate data.

# Acknowledgements

This work was performed using HPC resources from GENCI–IDRIS (Grant 2023-103245). This work was partially supported by Hi! Paris through the PhD funding of Nina Vesseron. The authors would like to thank Nicolas Papadakis for his valuable advice on leveraging the conjugate moment measure factorization for image restoration.

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

# A  Appendix / Supplementary Material

## A.1  Sketch of Proof for Theorem 1

*Proof Sketch.* The complete proof, along with the theorems used in the proof, can be found in Appendix A.2. We rely on the identity $\nabla w^* \circ \nabla w = \mathrm{id}$ that holds for strictly convex functions $w$ and show the existence of a strictly convex potential $w$ that verifies $\nabla w \sharp \rho = \mathfrak{P}_w$. To proceed, we first introduce some notations. Let $\mathcal{L}(\mathbb{R}^d)$ be the set of functions mapping $\mathbb{R}^d$ to $\mathbb{R} \cup \{+\infty\}$. Given a non-negligible set $\Omega \subset \mathbb{R}^d$ and a continuous function $v : \Omega \to \mathbb{R}$ such that $\int_{\mathbb{R}^d} e^{-v(z)} \mathrm{d}z$ is finite, we denote by $\mathfrak{P}_v^\Omega$ the probability measure with density

$$\frac{e^{-v(x)} 1_{x \in \Omega}}{\int_\Omega e^{-v(z)} \mathrm{d}z} \, .$$

We then define, for a probability $\rho$ supported on a non-negligible set $\Omega$, the map $G_\rho^\Omega : \mathcal{L}(\mathbb{R}^d) \to \mathcal{L}(\mathbb{R}^d)$ which to a potential $v$ associates the Brenier potential, defined in (4), from $\rho$ to $\mathfrak{P}_v^\Omega$

$$G_\rho^\Omega(v) := \mathcal{B}(\rho, \mathfrak{P}_v^\Omega)$$

The fixed points of $G_\rho^\Omega$ are precisely the convex functions $v$ such that $\nabla v \sharp \rho = \mathfrak{P}_v^\Omega$ from which we can construct $w$ defined as $w(x) = v(x)$ for $x \in \Omega$ and $w(x) = +\infty$ elsewhere. This function $w$ is convex and verifies $\nabla w \sharp \rho = \mathfrak{P}_w$ (since $\rho$ is supported on $\Omega$). We rely on Schauder's fixed point theorem to show that $G_\rho^\Omega$ admits a fixed-point. Given a Banach space $(X, \|.\|)$ and a compact, convex and nonempty set $\mathcal{M} \subset X$, Schauder's theorem states that any continuous operator $A : \mathcal{M} \to \mathcal{M}$ has at least one fixed point. In this proof, we consider the Banach space of continuous functions over a compact set $\Omega$, denoted $\mathscr{C}(\Omega)$, equipped with the supremum norm, and establish the following theorem that directly implies Theorem 1.

**Lemma 1.** *Let $\rho$ be an absolutely continuous probability measure supported on a compact, convex set $\Omega \subset \mathbb{R}^d$. Then $G_\rho^\Omega$ admits a fixed point in*

$$\mathcal{M} = \{f \in \mathscr{C}(\Omega) \text{ such that } \forall x, y \in \Omega, \ |f(x) - f(y)| \le R \|x - y\|_2 \text{ and } f(0_{\mathbb{R}^d}) = 0\}$$

*where $R$ is the radius of an euclidean ball that contains $\Omega$ i.e., $\Omega \subset B(0, R) = \{x \in \mathbb{R}^d, \|x\|_2 \le R\}$.*

We first show that the set $\mathcal{M}$ is a non-empty, compact, convex set of $X = (\mathscr{C}(\Omega), \| \ \|_\infty)$ by relying Arzela-Ascoli's theorem for the compactness. We then use Brenier's theorem that, given the absolute continuity of $\rho$, garantees the existence of the Brenier potential $\mathcal{B}(\rho, \mathfrak{P}_v^\Omega)$ and the fact that $G_\rho^\Omega$ is therefore well-defined on the set $\mathcal{M} \subset \mathscr{C}(\Omega)$. Moreover, the gradient of the obtained potential $G_\rho^\Omega(v)$ transports $\rho$ on $\mathfrak{P}_v^\Omega$ which is compactly supported on $\Omega \subset B(0, R)$. For this reason, $\nabla G_\rho^\Omega(v)$ is bounded by $R$ on $\Omega$ and the obtained potential is Lipschitz continuous with constant $R$. This permits to show that $G_\rho^\Omega(\mathcal{M}) \subseteq \mathcal{M}$. To prove the continuity of $G_\rho^\Omega$ on $\mathcal{M}$, one can remark that $G_\rho^\Omega = H_\rho \circ F^\Omega$, with $F^\Omega(v) = \mathfrak{P}_v^\Omega$ and $H_\rho(\mu) = \mathcal{B}(\rho, \mu)$, both viewed as functions from $\mathscr{C}(\Omega)$ to $\mathscr{C}(\Omega)$. To show that $F^\Omega$ is continuous, we use the definition of continuity in $X = (\mathscr{C}(\Omega), \| \ \|_\infty)$ while we rely on Theorem 1.52 from Santambrogio [2015] to prove that $H_\rho$ is continuous. We conclude by applying Schauder's theorem.

## A.2  Proof Lemma 1

Let $\rho$ be an absolutely continuous probability measure supported on a compact, convex set $\Omega \subset \mathbb{R}^d$. $\Omega$ being compact, it is bounded and there exists a real number $R$ such that $\Omega$ be included in the ball $B(0, R)$. We begin by recalling Schauder's fixed point theorem, which is central to our proof for showing that $G_\rho^\Omega$, defined as:

$$G_\rho^\Omega(v) := \mathcal{B}(\rho, \mathfrak{P}_v^\Omega)$$

where $\mathfrak{P}_v^\Omega$ the probability measure with density

$$\frac{e^{-v(x)} 1_{x \in \Omega}}{\int_\Omega e^{-v(z)} \mathrm{d}z} \, ,$$

admits a fixed-point.

**Theorem** (Schauder's fixed point Theorem). *Let $(X, \|.\|)$ be a Banach space and $\mathcal{M} \subset X$ is compact, convex, and nonempty. Any continuous operator $A : \mathcal{M} \to \mathcal{M}$ has at least one fixed point.*

We consider the Banach space $X = (\mathscr{C}(\Omega), \| \ \|_\infty)$ to study the continuity of $G_\rho^\Omega$ on the set

$$\mathcal{M} = \{f \in \mathscr{C}(\Omega) \text{ such that } \forall x, y \in \Omega, |f(x) - f(y)| \leq R\|x - y\|_2 \text{ and } f(0_{\mathbb{R}^d}) = 0\}.$$

We prove the two following lemmas in order to use Schauder's theorem.

**Lemma 2.** *The set $\mathcal{M}$ is a non-empty, compact, convex set of $X$.*

**Lemma 3.** *$G_\rho^\Omega$ is well defined and continuous on $\mathcal{M}$ and $G_\rho^\Omega(\mathcal{M}) \subseteq \mathcal{M}$.*

By applying these two lemmas along with Schauder's theorem, we conclude that $G_\rho^\Omega$ has a fixed point $v_{\text{opt}}$ in $M \subset \mathscr{C}(\Omega)$. Moreover, the absolute continuity of $\mathfrak{P}_{v_{\text{opt}}}^\Omega$ ensures that the optimal solution $v_{\text{opt}}$ whose gradient transports $\rho$ onto $\mathfrak{P}_{v_{\text{opt}}}^\Omega$ is strictly convex on the support of $\rho$, $\Omega$. Additionally, the gradient of its Legendre transform $\nabla v_{\text{opt}}^*$ is the OT map from $\mathfrak{P}_{v_{\text{opt}}}^\Omega$ to $\rho$.

### A.2.1 Proof of Lemma 2.

$\mathcal{M}$ **is non-empty**    0 is included in $\mathcal{M}$ which is therefore not empty.

$\mathcal{M}$ **is convex**    Let $f$ and $g$ functions of $\mathcal{M}$ and $\lambda \in [0, 1]$. We have that $\lambda f + (1 - \lambda)g \in \mathscr{C}(\Omega)$ and that $\lambda f(0) + (1 - \lambda)g(0) = 0$. Let $x \in \Omega$ and $y \in \Omega$, then :

$$|(\lambda f(x) + (1 - \lambda)g(x)) - (\lambda f(y) + (1 - \lambda)g(y))| \leq \lambda|f(x) - f(y)| + (1 - \lambda)|g(x) - g(y)|$$
$$\leq \lambda R\|x - y\|_2 + (1 - \lambda)R\|x - y\|_2 = R\|x - y\|_2$$

where, for the first inequality, we rely on the fact that the absolute value verifies the triangular inequality and for the second inequality, we use that $f$ and $g$ belong to $\mathcal{M}$ and are therefore $R$-lipschtiz.

$\mathcal{M}$ **is compact**    We rely on Arzela-Ascoli theorem which is recalled below to show that the set $\mathcal{M}$ is compact. We also recall the definitions of equicontinuity and equiboundness.

**Definition.** *A family of functions $\mathcal{F} \subset \mathscr{C}(\Omega)$ is equibounded if there exists a constant $C_0$ such that for all $x \in \Omega$ and $f \in \mathcal{F}$, we have $|f(x)| \leq C_0$.*

**Definition.** *A family of functions $\mathcal{F} \subset \mathscr{C}(\Omega)$ is equicontinous if for all $\varepsilon > 0$, there exists $\delta$ such that for $x, y \in \Omega$,*

$$\|x - y\| < \delta \implies \forall f \in \mathcal{F} \ |f(x) - f(y)| < \varepsilon$$

**Theorem** (Arzela-Ascoli). *Let $(K, d)$ a compact space, a family of functions $\mathcal{F} \subset \mathscr{C}(K)$ is relatively compact if and only if $\mathcal{F}$ is equibounded and equicontinuous.*

The family of functions $\mathcal{M}$ is equibounded. In fact, for $f \in M$ and $x \in \Omega$, we have that:

$$|f(x)| = |f(x) - f(0)| \leq R\|x - 0\|_2 \leq R^2$$

The first equality comes from the fact that $f \in M$, therefore $f(0) = 0$, the first inequality uses the fact that $f$ is Lipschitz continuous with constant $R$ while the last one is due to the fact that $x \in \Omega \subset B(0, R)$.

The family of functions $\mathcal{M}$ is equicontinuous. For $\varepsilon > 0$, we define $\delta = \frac{\varepsilon}{R}$. Let $f \in \mathcal{M}$ and $x, y \in \Omega$ such that $\|x - y\|_2 < \delta$ then because $f$ is Lipschitz on $\Omega$, we have that

$$|f(x) - f(y)| \leq R\|x - y\|_2 < R\delta = \varepsilon .$$

Then, according to Arzela-Ascoli theorem, $\mathcal{M}$ is relatively compact i.e. its closure is compact.

$\mathcal{M}$ **is closed in** $X = (\mathscr{C}(\Omega), \| \ \|_\infty)$    Let $(f_n)$ be a sequence of functions in $\mathcal{M}$ that converges uniformly to a limiting function $f$. We have that $f \in \mathscr{C}(\Omega)$ which is closed. Moreover, because $(f_n)$ converges uniformly to $f$, it also converges pointwise and in particular $f_n(0) = 0 \to 0 = f(0)$. Finally, because $\forall n$, $f_n$ is Lipschitz continuous, we have

$$|f_n(x) - f_n(y)| \leq R\|x - y\|_2 \ \forall n$$

By taking the limit when $n \to +\infty$ we get that:

$$|f(x) - f(y)| \leq R\|x - y\|_2$$

and $f$ is Lipschitz continuous with constant $R$ and $f \in \mathcal{M}$. This proves that $\mathcal{M}$ is closed and that $\mathcal{M}$ is therefore compact.

### A.2.2   Proof of Lemma 3.

$G_\rho^\Omega$ **is well-defined on** $\mathcal{M}$   For a given $v \in \mathcal{M}$, the fact that $v \in \mathscr{C}(\Omega)$ and that $\Omega$ is non-negligible (because $\rho$ is an absolutely continuous probability measure supported on $\Omega$) ensure the measure $\mathfrak{P}_v^\Omega$ to be well-defined. Using Brenier's theorem, the absolute continuity of $\rho$ garantees the existence of the Brenier potential $\mathcal{B}(\rho, \mathfrak{P}_v^\Omega)$ and the fact that $G_\rho^\Omega$ is therefore well-defined on the set $\mathcal{M} \subset \mathscr{C}(\Omega)$.

$G_\rho^\Omega(\mathcal{M}) \subset \mathcal{M}$   For a given $v \in \mathcal{M}$, the Brenier potential $G_\rho^\Omega(v)$ verifies $G_\rho^\Omega(v)(0) = 0$ by definition. As a convex function which takes real values on $\Omega$, $G_\rho^\Omega(v)$ is continuous on $\Omega$ and $G_\rho^\Omega(v) \in \mathscr{C}(\Omega)$. Moreover, the gradient of the obtained potential $\nabla G_\rho^\Omega(v)$ transports $\rho$ on $\mathfrak{P}_v^\Omega$. The probability measures $\mathfrak{P}_v^\Omega$ and $\rho$ are both compactly supported on $\Omega \subset B(0, R)$ which implies that the support of the optimal transport plan $\gamma = (i_d, \nabla G_\rho^\Omega(v))_\# \rho$ is included in $\Omega \times \Omega$. $\Omega$ being the support of $\rho$, we have that for each $x \in \Omega$, $\nabla G_\rho^\Omega(v)(x) \in \Omega \subset B(0, R)$ and in particular $\|\nabla G_\rho^\Omega(v)(x)\|_2 \leq R$. $\Omega$ being convex and $G_\rho^\Omega(v)$ being differentiable in the interior of $\Omega$, we can apply the mean value theorem to deduce that:

$$\forall x, y \in \Omega \; |G_\rho^\Omega(v)(x) - G_\rho^\Omega(v)(y)| \leq \sup_{z \in \Omega} \|\nabla G_\rho^\Omega(v)(z)\|_2 \|x - y\|_2 \leq R\|x - y\|_2$$

$G_\rho^\Omega(v)$ is therefore Lipschitz continuous with constant $R$ and it belongs to the set $\mathcal{M}$.

$G_\rho^\Omega$ **is continuous on** $\mathcal{M}$   To prove the continuity of $G_\rho^\Omega$ on $\mathcal{M}$, one can remark that $G_\rho^\Omega = H_\rho \circ F^\Omega$, with

$$F^\Omega : \; \mathcal{M} \longrightarrow \mathcal{N}$$
$$v \longrightarrow \mathfrak{P}_v^\Omega$$

and

$$H_\rho : \; \mathcal{N} \longrightarrow \mathscr{C}(\Omega)$$
$$\mu \longrightarrow \mathcal{B}(\rho, \mu)$$

with $\mathcal{N} = \{\mathfrak{P}_v^\Omega, v \in \mathscr{C}(\Omega)\} \subset \mathscr{C}(\Omega)$. We show that both function $F^\Omega$ and $H_\rho$ are continuous in $X = (\mathscr{C}(\Omega), \|\ \|_\infty)$. Let $v \in \mathscr{C}(\Omega)$ and $(v_n)$ a familly of functions of $\mathscr{C}(\Omega)$ that converges uniformly to $v$. We will show that the family $(F^\Omega(v_n))$ converges uniformly to $F^\Omega(v)$. Let $x$ in $\Omega$ and $n \in \mathbb{N}$, we have:

$$|F^\Omega(v_n)(x) - F^\Omega(v)(x)| = \left| \frac{e^{-v_n(x)}}{\int_\Omega e^{-v_n(z)} \mathrm{d}z} - \frac{e^{-v(x)}}{\int_\Omega e^{-v(z)} \mathrm{d}z} \right|$$
$$= \left| \frac{\left(e^{-v_n(x)} \int_\Omega e^{-v(z)} \mathrm{d}z\right) - \left(e^{-v(x)} \int_\Omega e^{-v_n(z)} \mathrm{d}z\right)}{\int_\Omega e^{-v_n(z)} \mathrm{d}z \int_\Omega e^{-v(z)} \mathrm{d}z} \right|$$

We will use the following lemma:

**Lemma 4.** *If $(v_n)$ converges uniformly to $v$ in $\mathscr{C}(\Omega)$ with $\Omega$ compact then $e^{-v_n}$ converges uniformly to $e^{-v}$ in $\mathscr{C}(\Omega)$.*

*Proof of Lemma 4.* Let us denote by $f$ the real function $f : x \longrightarrow e^{-x}$. $f$ is continuous on $\mathbb{R}$. Because $v \in \mathscr{C}(\Omega)$ and $\Omega$ compact, we have, by Heine's theorem, that $v(\Omega) \subset \mathbb{R}$ is compact and therefore bounded. There exists $a$ and $b$ such that $v(\Omega) \subseteq [a, b]$. Moreover because $(v_n)$ converges uniformly to $v$, there exists $N$ such that $n \geq N \implies v_n(\Omega) \subseteq [a - 1, b + 1]$. $f$ being continuous on $\mathbb{R}$, it is uniformly continuous on the interval $[a - 1, b + 1]$ according to Heine's theorem. We deduce that $(f(v_n))$ converges uniformly to $f(v)$. $\qquad \square$

On the other hand, the function $g : t \longrightarrow e^{-v(t)}$ is continuous on the compact $\Omega$ as a composition of continuous functions. By Heine's theorem, the image of $\Omega$ by $g$ is a compact and there exists $c, \ell \in \mathbb{R}^{+*}$ such that $g(\Omega) \subset [c, \ell]$ with $c > 0$. Because $e^{-v_n}$ converges uniformly to $e^{-v}$ in $\mathscr{C}(\Omega)$, there exists $N_0$ such that $n \geq N_0 \implies e^{-v_n}(z) \geq \frac{c}{2} \; \forall z \in \Omega$. As a consequence, we have for $n \geq N_0$:

$$\int_\Omega e^{-v_n(z)}\mathrm{d}z \int_\Omega e^{-v(z)}\mathrm{d}z \geq \int_\Omega \frac{c}{2}\mathrm{d}z \int_\Omega c\,\mathrm{d}z = \frac{c^2}{2}|\Omega|^2$$

with $|\Omega|$ the lebesgue measure of the compact $\Omega$. $\Omega$ being non-negligible, we have $|\Omega| > 0$. Let $C_0 = \frac{2}{c^2|\Omega|^2}$, we have shown that for $n \geq N_0$ we had:

$$\left| \frac{1}{\int_\Omega e^{-v_n(z)}\mathrm{d}z \int_\Omega e^{-v(z)}\mathrm{d}z} \right| < C_0$$

Furthermore, one notes that:

$$\left| \left( e^{-v_n(x)} \int_\Omega e^{-v(z)}\mathrm{d}z \right) - \left( e^{-v(x)} \int_\Omega e^{-v_n(z)}\mathrm{d}z \right) \right|$$

$$= \left| \left( e^{-v_n(x)} \int_\Omega e^{-v(z)}\mathrm{d}z \right) - \left( e^{-v(x)} \int_\Omega e^{-v(z)}\mathrm{d}z \right) + \left( e^{-v(x)} \int_\Omega e^{-v(z)}\mathrm{d}z \right) - \left( e^{-v(x)} \int_\Omega e^{-v_n(z)}\mathrm{d}z \right) \right|$$

$$\leq \left| \left( e^{-v_n(x)} - e^{-v(x)} \right) \int_\Omega e^{-v(z)}\mathrm{d}z \right| + \left| e^{-v(x)} \left( \int_\Omega e^{-v(z)}\mathrm{d}z - \int_\Omega e^{-v_n(z)}\mathrm{d}z \right) \right|$$

$$\leq \left| e^{-v_n(x)} - e^{-v(x)} \right| \ell |\Omega| + \ell \left| \int_\Omega e^{-v(z)}\mathrm{d}z - \int_\Omega e^{-v_n(z)}\mathrm{d}z \right|$$

$$\leq \left\| e^{-v_n} - e^{-v} \right\|_\infty \ell |\Omega| + \ell \int_\Omega \left| e^{-v(z)} - e^{-v_n(z)} \right| \mathrm{d}z$$

$$\leq 2\ell|\Omega| \left\| e^{-v_n} - e^{-v} \right\|_\infty$$

Until now, we have shown that for $n \geq N_0$ we had, for $x \in \Omega$:

$$|F^\Omega(v_n)(x) - F^\Omega(v)(x)| \leq C_0 2\ell|\Omega| \left\| e^{-v_n} - e^{-v} \right\|_\infty$$

Let $\varepsilon > 0$, let us define $\varepsilon_0 = \frac{\varepsilon}{C_0 2\ell|\Omega|} > 0$. Because $(e^{-v_n})$ converges uniformly to $e^{-v}$, there exists $N_1$ such that $n \geq N_1 \implies \left\| e^{-v_n} - e^{-v} \right\|_\infty < \varepsilon_0$. Let us denote $N = \max(N_0, N_1)$, we have that for $n \geq N$,

$$|F^\Omega(v_n)(x) - F^\Omega(v)(x)| \leq \varepsilon$$

Because this is true for any $x \in \Omega$, we have $\left\| F^\Omega(v_n) - F^\Omega(v) \right\|_\infty \leq \varepsilon$. We just proved that $F^\Omega$ is continuous at $v$ for any $v \in \mathscr{C}(\Omega)$ ie $F^\Omega$ is continuous on $\mathscr{C}(\Omega)$ and in particular it is continuous on $\mathcal{M} \subset \mathscr{C}(\Omega)$.

We will now show that $H_\rho$ is continuous on $\mathcal{N}$. Let $\mu \in \mathcal{N}$ and let $(\mu_n)$ a family of densities from $\mathcal{N}$ that converges uniformly to $\mu$. For $n \in \mathbb{N}$, we denote by $\mathcal{P}(\mu_n)$ and $\mathcal{P}(\mu)$ the probability measures associated to the respective densities $\mu_n$ and $\mu$. Because $(\mu_n)$ converges uniformly to $\mu$, we have that $\mathcal{P}(\mu_n)$ converges in distribution to $\mathcal{P}(\mu)$. Then according to Santambrogio [2015] (Theorem 1.52), the family of Kantorovich potentials from $\rho$ to $\mathcal{P}(\mu_n)$ with value 0 in 0 that we denote $(f_n^\star)$ converges uniformly to the Kantorovich potential $f^\star$ from $\rho$ to $\mathcal{P}(\mu)$ whose value is 0 in 0. Therefore, the family of Brenier potentials $(H_\rho(\mu_n) : t \to \frac{1}{2}\|t\|^2 - f_n^\star(t))$ converges uniformly to $H_\rho(\mu) = \frac{1}{2}\| \cdot \|^2 - f^\star$. We conclude that $H_\rho$ is continuous on $\mathcal{N}$ and by composition of continuous functions $G_\rho^\Omega$ is continuous on $\mathcal{M}$.

### A.3 Additional Proof for Proposition 1

Let us observe that $\rho$ is the moment measure of $u$ if and only if $\nabla u$ is the Monge map from $\mathfrak{P}_u$ to $\rho$. One can then consider quadratic functions $u_{r,\Delta}(x) = \frac{1}{2}(x-r)^T\Delta^{-1}(x-r)$ whose Gibbs distribution is $\mathfrak{P}_u = \mathcal{N}(r,\Delta)$ and use the expression [Gelbrich, 1990] of the OT map between two Gaussian distributions $\mathcal{N}(m_A,\Sigma_A)$ and $\mathcal{N}(m_B,\Sigma_B)$:

$$T_{m_A,\Sigma_A,m_B,\Sigma_B} : x \mapsto m_B + M(x - m_A)$$

where $M = \Sigma_A^{-1/2}(\Sigma_A^{1/2}\Sigma_B\Sigma_A^{1/2})^{1/2}\Sigma_A^{-1/2}$, to solve the equation

$$\nabla u_{r,\Delta}(x) = T_{r,\Delta,0,\Sigma}(x) \ \ \forall x \in \mathbb{R}^d$$

One get:

$$\Delta^{-1}(x-r) = M(x-r) \ \ \text{with } M = \Delta^{-1/2}(\Delta^{1/2}\Sigma\Delta^{1/2})^{1/2}\Delta^{-1/2}$$

By identification, we obtain that $\Delta^{-1} = M = \Delta^{-1/2}(\Delta^{1/2}\Sigma\Delta^{1/2})^{1/2}\Delta^{-1/2}$ which leads to $\Delta = \Sigma^{-1}$. It means that the functions in $\{u_{r,\Sigma^{-1}}, r \in \mathbb{R}^d\}$ are fixed-point of $G_\rho$ with $\rho = \mathcal{N}(0,\Sigma)$. According to Cordero-Erausquin and Klartag [2015], the moment potential is unique up to translation, which implies that there are no other convex functions whose moment measure is $\rho$.

### A.4 Proof of Proposition 2

The proof is similar to the additional proof of Proposition 1 provided Section A.3. Given $\rho = \mathcal{N}(m,\Sigma)$, we are looking for $w$ such that $\nabla w^*\#\mathfrak{P}_w = \rho$ and $\mathfrak{P}_w = \mathcal{N}(r,\Delta)$. This means that $w(x) = \frac{1}{2}(x-r)^T\Delta^{-1}(x-r)$ and $w^*(x) = \frac{1}{2}x^T\Delta x + x^T r$. Moreover $\nabla w^* : x \to \Delta x + r$ must be the OT map between $\mathfrak{P}_w = \mathcal{N}(r,\Delta)$ and $\rho = \mathcal{N}(m,\Sigma)$. The OT map between these two gaussians is known in closed form and is given by $x \to m + M(x-r)$ with $M = \Delta^{-1/2}(\Delta^{1/2}\Sigma\Delta^{1/2})^{1/2}\Delta^{-1/2}$. Solving $\Delta = \Delta^{-1/2}(\Delta^{1/2}\Sigma\Delta^{1/2})^{1/2}\Delta^{-1/2}$ gives $\Delta = \Sigma^{1/3}$ while solving $r = m - Mr$ gives $r = (I_d + \Sigma^{1/3})^{-1}m$. We conclude that $\rho$ is the conjugate moment measure of the unique potential $w$ among those whose Gibbs distribution is Gaussian.

## B Experiments in the 1D case where $\rho$ is known from samples

### B.1 Proposed algorithm for moment measures

For a probability measure $\rho \in \mathcal{P}(\mathbb{R}^d)$, we define the map $J_\rho : \mathcal{L}(\mathbb{R}^d) \to \mathcal{L}(\mathbb{R}^d)$ which to a potential $u$ associates the Brenier potential from $\mathfrak{P}_u$ to $\rho$

$$J_\rho(u) := \mathcal{B}(\mathfrak{P}_u, \rho)$$

The fixed point of $J_\rho$ correspond exactly to the moment potentials of $\rho$. This observation motivates the following fixed-point iteration scheme to compute a moment potential of $\rho$:

$$u_0 := \tfrac{1}{2}\|\cdot\|^2; \quad \forall t \geq 1, \quad u_{t+1} := J_\rho(u_t), \tag{11}$$

In the 1D case, the OT map between density functions $\mu$ and $\nu$ is known in closed form (see §4.2) which enables the exact application of the fixed-point algorithm.

### B.2 Details on the 1D experiments

We conducted two experiments, with $\rho$ defined as a mixture of Gaussian distributions. In the first experiment, $\rho$ is a mixture of four Gaussians: $\mathcal{N}(-0.1, 0.07), \mathcal{N}(0.3, 0.1), \mathcal{N}(0.3, 0.1), \mathcal{N}(0.7, 0.15)$ with mixture weights $\frac{1}{7}, \frac{3}{7}, \frac{2}{7}, \frac{1}{7}$, respectively. In the second experiment, $\rho$ is a mixture of the three Gaussians $\mathcal{N}(-0.8, 0.4), \mathcal{N}(1.5, 0.6), \mathcal{N}(3, 0.5)$, with weights $\frac{1}{2}, \frac{1}{3}, \frac{1}{6}$. To approximate these mixtures, we used 400,000 samples and estimated their histograms using 100,000 bins. This allowed us to compute the cumulative distribution and quantile functions using `numpy`'s `cumsum` and `quantile` functions, which were essential for estimating the OT map at each iteration of the fixed-point algorithm. The algorithm converged after 300 iterations.

### B.3 Code for the 1D experiments

**Code for estimating the moment potential in 1D**

Listing 1: Code for estimating the moment potential in 1D

```
@jax.jit
def compute_ot_map(positions_source, freq_source, samples_target):
  cdf = jnp.cumsum(freq_source, axis=0)
  cdf = cdf/cdf[-1]
  quantile_fn = jax.vmap(lambda x: jnp.quantile(samples_target, x))
  inverse_cdf = quantile_fn(cdf)
  return (positions_source, inverse_cdf, freq_source)

def compute_next_measure(positions_source, freq_source, samples_target):
  positions_source, inverse_cdf, freq_source = compute_ot_map(positions_source,
      freq_source, samples_target)
  u = integrate.cumtrapz(inverse_cdf, positions_source, initial=0)
  weights = jnp.array(scipy.special.softmax(-u)) * jnp.sum(freq_source)
  return (positions_source, weights), (u, inverse_cdf)

# hyperparameters
nb_bins = 100000
bound_min = -11
bound_max = 11
nb_points = 400000
nb_bins_plot = 1000

# generate source and target distributions
rng = jax.random.PRNGKey(0)
rng_source, rng_target, rng = jax.random.split(rng, 3)

mu_target = jnp.array([-0.6, -0.1, 0.3, 0.7]) * 0.8
sigma_target = jnp.array([0.15, 0.07, 0.1, 0.15]) * 0.8
alpha_target = jnp.array((1,3,2,1))
sampler_target = multivariate_gaussians_1D(mu=mu_target, sigma=sigma_target, alpha=
    alpha_target)
samples_target = sampler_target.generate_samples(rng_target, nb_points)
samples_target = samples_target - jnp.mean(samples_target)

samples_source = jax.random.normal(rng_source, shape=(nb_points,))

# construct bins and histogram
freq_source, edges_source = jnp.histogram(samples_source, nb_bins, range=(bound_min,
    bound_max))
positions_source = (edges_source[:-1] + edges_source[1:]) / 2.0
freq_target, edges_target = jnp.histogram(samples_target, nb_bins, range=(bound_min,
    bound_max))
positions_target = (edges_target[:-1] + edges_target[1:]) / 2.0

for i in range(300):
  (positions_source, freq_source), (u, grad_u) = compute_next_measure(
      positions_source, freq_source, samples_target)
```

**Code for estimating the *conjugate* moment potential in 1D: CMFGen**

Listing 2: Code for estimating the conjugate moment potential in 1D

```python
@jax.jit
def compute_ot_map_star(positions_target, freq_target, samples_source):
    cdf = jnp.cumsum(freq_target, axis=0)
    cdf = cdf/cdf[-1]
    quantile_fn = jax.vmap(lambda x: jnp.quantile(samples_source, x))
    inverse_cdf = quantile_fn(cdf)
    return (positions_target, inverse_cdf, freq_target)

def compute_next_measure(positions_target, freq_target, samples_source):
    positions_target, inverse_cdf, freq_target = compute_ot_map_star(positions_target
        , freq_target, samples_source)
    u = integrate.cumtrapz(inverse_cdf, positions_target, initial=0)
    weights = jnp.array(scipy.special.softmax(-u)) * jnp.sum(freq_target)
    return (positions_target, weights), (u, inverse_cdf)

# hyperparameters
nb_bins = 100000
bound_min = -4
bound_max = 4
nb_points = 400000
nb_bins_plot = 1000

# generate source and target distributions
rng = jax.random.PRNGKey(0)
rng_source, rng_target, rng = jax.random.split(rng, 3)

mu_target = jnp.array([-0.6, -0.1, 0.3, 0.7]) * 0.8
sigma_target = jnp.array([0.15, 0.07, 0.1, 0.15]) * 0.8
alpha_target = jnp.array((1,3,2,1))
sampler_target = multivariate_gaussians_1D(mu=mu_target, sigma=sigma_target, alpha=
    alpha_target)
samples_target = sampler_target.generate_samples(rng_target, nb_points)
samples_target = samples_target - jnp.mean(samples_target)

samples_source = jax.random.normal(rng_source, shape=(nb_points,))

# construct bins and histogram
freq_source, edges_source = jnp.histogram(samples_source, nb_bins, range=(bound_min,
     bound_max))
positions_source = (edges_source[:-1] + edges_source[1:]) / 2.0

freq_source_plot, edges_source_plot = jnp.histogram(samples_source, nb_bins)
positions_source_plot = (edges_source_plot[:-1] + edges_source_plot[1:]) / 2.0

freq_target, edges_target = jnp.histogram(samples_target, nb_bins, range=(bound_min,
     bound_max))
positions_target = (edges_target[:-1] + edges_target[1:]) / 2.0

for i in range(300):
    rng_, rng = jax.random.split(rng, 2)
    (positions_source, freq_source), (u_star, grad_u_star) = compute_next_measure(
        positions_target, freq_target, samples_source)
```

```
samples_source = jax.random.choice(rng_, positions_source, shape=(nb_points,), p=
    freq_source)
samples_source = samples_source - jnp.mean(samples_source)
```

## B.4 Other figures

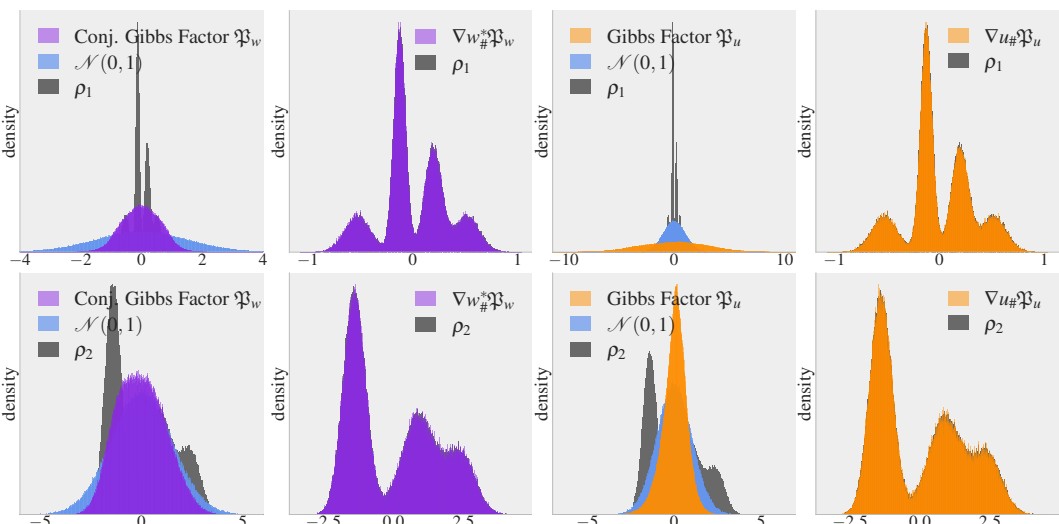

Figure 8: Comparison between the **Gibbs factor** $\mathfrak{P}_u$ and the **conjugate Gibbs factor** $\mathfrak{P}_w$ for two mixtures of 1D Gaussian distributions, $\rho_1$ and $\rho_2$. Pushforward densities $\nabla w^* \sharp \mathfrak{P}_w$ and $\nabla u \sharp \mathfrak{P}_u$ closely match the target distributions, illustrating that the fixed-point algorithms converged and succefully estimated the (conjugate) potentials.

## C CMFMA experiments

### C.1 1D experiments: Figures

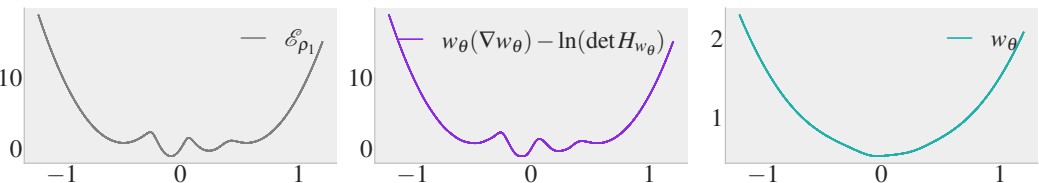

Figure 9: Energy associated to the mixture of gaussians $\rho_1$ (left), learned energy (middle), associated potential (right).

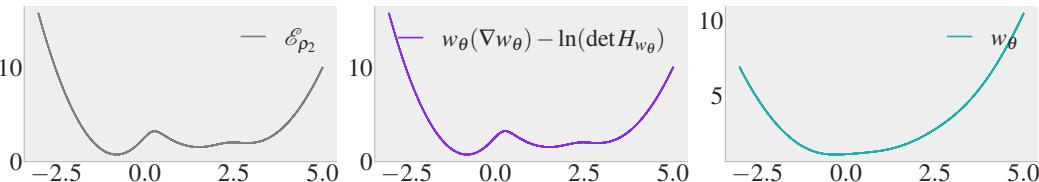

Figure 10: Energy associated to the mixture of gaussians $\rho_2$ (left), learned energy (middle), associated potential (right).

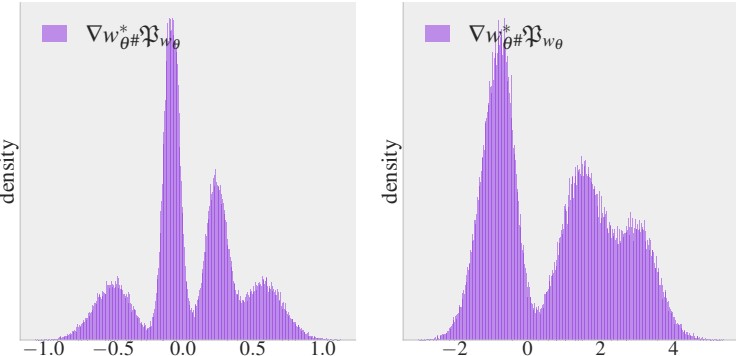

Figure 11: The pushforward densities $\nabla w_\theta^* \sharp \mathfrak{P}_{w_\theta}$ closely match the target distributions $\rho_1$ and $\rho_2$, illustrating that CMFMA succesfully estimated the (conjugate) potentials.

## C.2 2D experiments

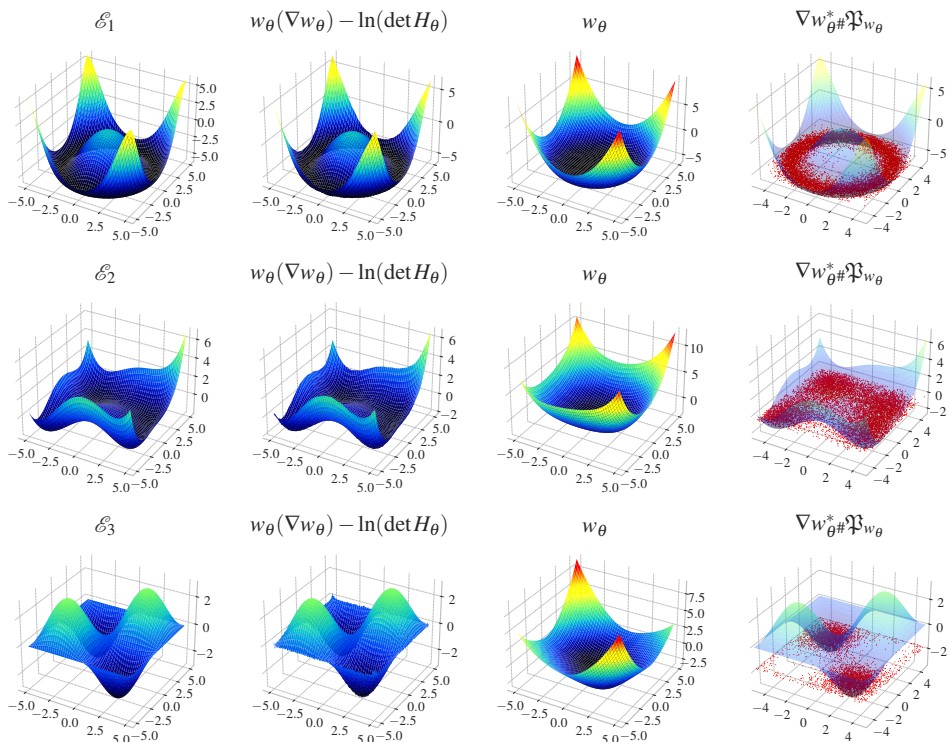

Figure 12: **Learning the conjugate moment potential from an energy**. $\mathcal{E}_1$, $\mathcal{E}_2$ and $\mathcal{E}_3$ are learned by regression with CMFSamp. The second column shows the learned energy, the third column displays the corresponding conjugate moment potential, and the fourth column presents samples (in red) drawn from $\nabla w_\theta^* \sharp \mathfrak{P}_{w_\theta}$.

The expressions of the three energies used for the 2D experiments with CMFMA are the following:

$$\mathcal{E}_1(x, y) = -6 \cdot \sin\left(\left(\tfrac{3}{10}x\right)^2 + \left(\tfrac{3}{10}y\right)^2\right)$$

$$\mathcal{E}_2(x, y) = \frac{(x^2 + y - 11)^2 + (x + y^2 - 7)^2}{100} - 2$$

$$\mathcal{E}_3(x, y) = 3 \cdot \cos\left(\tfrac{2\pi}{10}x - \tfrac{\pi}{2}\right) \cdot \cos\left(\tfrac{2\pi}{10}y - \tfrac{\pi}{2}\right)$$

# D CMFGen experiments

## D.1 2D experiments: comparison with generative ICNNs

In this experiment, we compare our method, CMFGen, with the standard ICNN-based generative pipeline (see Amos [2023]) that maps a standard gaussian noise to data. The two approaches are evaluated using the Sinkhorn divergence between generated samples and reference samples from the target distribution, computed over batches of size $N = 2048$. We repeat this evaluation across 100 random seeds applied to both generated and target distributions, and the resulting divergences are summarized by the boxplots in Figure 13. As a baseline, the leftmost column reports the 2-Sinkhorn divergence between two independent samples drawn from the target distribution. The $\varepsilon$ parameter used to compute the Sinkhorn divergence is adapted to the scale of the data. As in Vesseron and Cuturi [2024], we set $\varepsilon = 0.05 \cdot \mathbb{E}_{x,x' \sim \rho}|x - x'|^2$, where the expectation is estimated from two batches of 2048 samples each, drawn independently from the target distribution $\rho$.

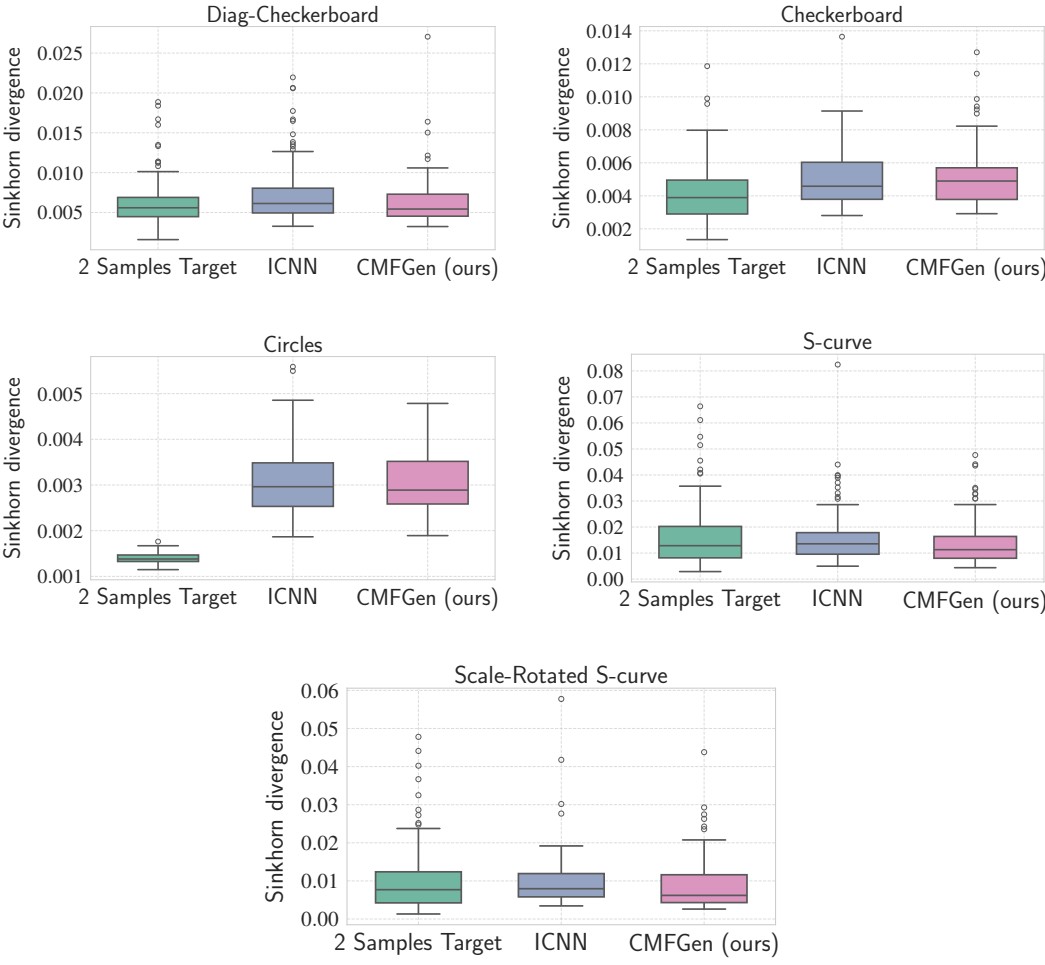

Figure 13: Comparison with generative ICNN across various datasets: Diagonal Checkerboard, Checkerboard, Circles, S-curve, and Scale-Rotated S-curve. Each boxplot shows the 2-Sinkhorn divergences between generated and target samples, computed over $N = 2048$ samples and averaged over 100 random seeds. For each dataset, the first boxplot shows the divergence between two independent samples from the target, the second for the generative ICNN, and the third for our method, CMFGen.

**D.2   Cartoon dataset: comparison with generative ICNNs**

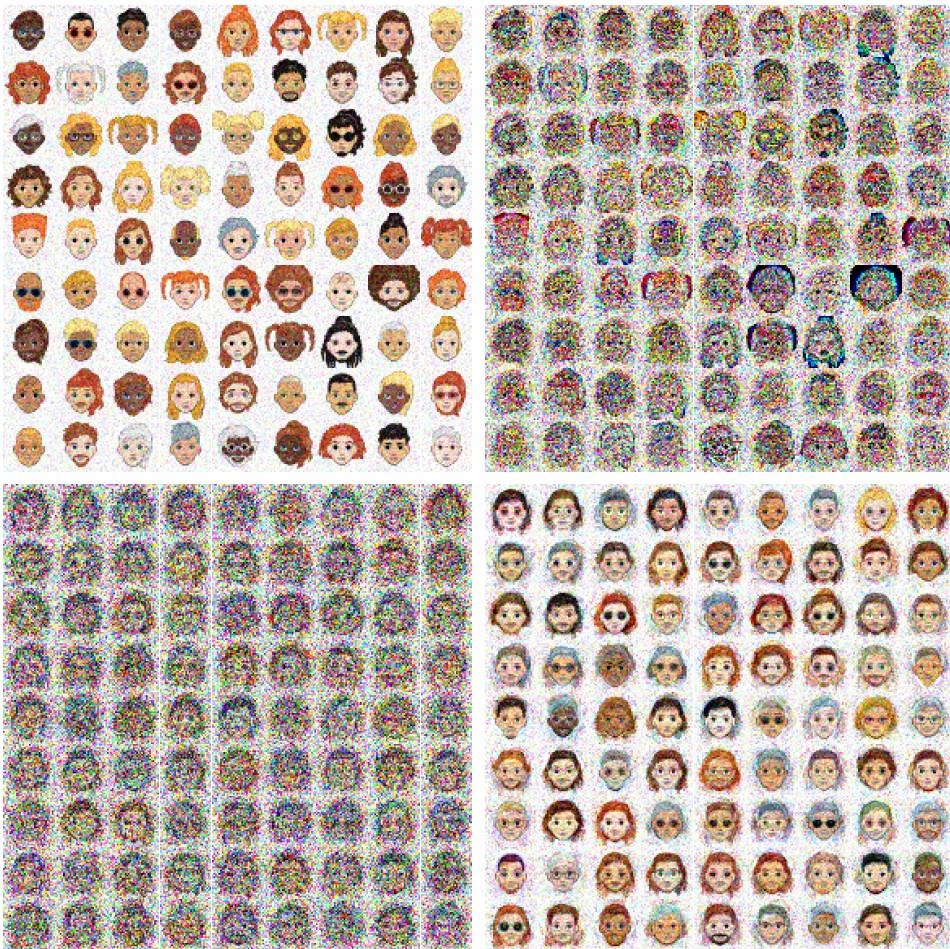

Figure 14: Additional images for the cartoon experiment using CMFGen. Top left: Images from the cartoon dataset; top right: Transported images using the map $\nabla w$; bottom left: Samples from the Gibbs distribution $\mathfrak{P}_{w_\theta}$; bottom right: Samples from the distribution $\nabla w_\theta^* \sharp \mathfrak{P}_{w_\theta}$.

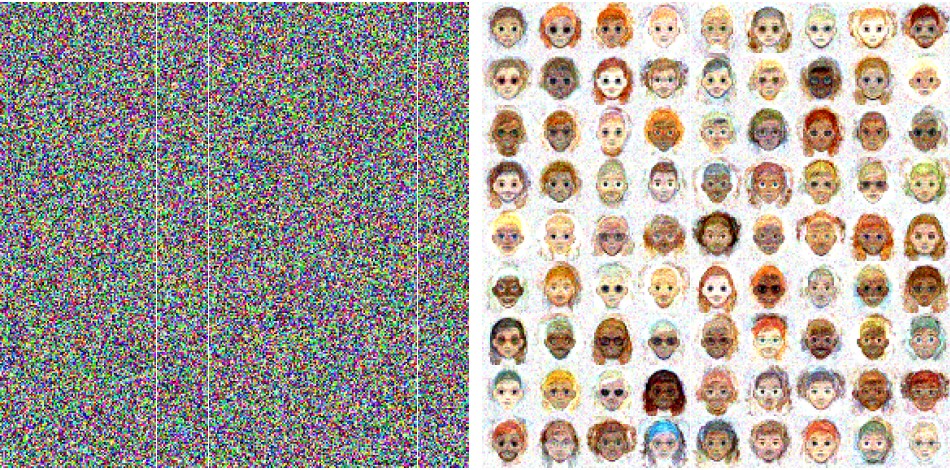

Figure 15: Images produced with a generative ICNN on the cartoon dataset. Left: Gaussian noise; Right: Generated samples by the ICNN with the same architecture as used by CMFGen, i.e., an ICNN with five hidden layers of size 512 and four quadratic input connections.

### D.3    Image Reconstruction: MNIST and Cartoon dataset

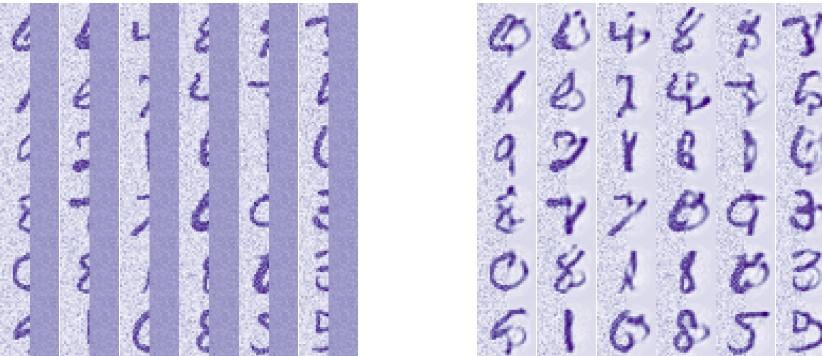

Figure 16: The learned $w_\theta$ trained on the MNIST dataset is used for a post-processing task to recover the masked pixels. Gradient ascent is performed on the masked pixels to maximize the log-probability of the full image. Left: Masked images; Right: Reconstructed images.

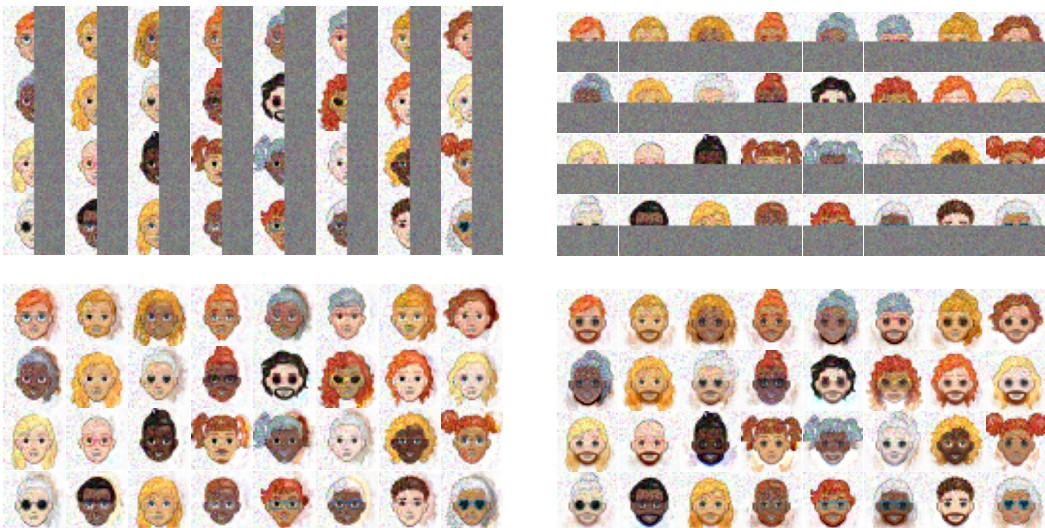

Figure 17: The learned $w_\theta$ trained on the Cartoon dataset is used for a post-processing task to recover the masked pixels. Gradient ascent is performed on the masked pixels to maximize the log-probability of the full image. Top: Masked images; Bottom: Reconstructed images.

## E    Hyperparameters and infrastructure

All runs have been made on a single GPU V100-32GB.

**Noisy MNIST and Cartoon Datasets**    For the MNIST experiment, We convolve the MNIST images with Gaussian noise of standard deviation $0.3$ to ensure the existence of the OT map from the MNIST distribution to the log-concave distribution $\mathfrak{P}_w$, as guaranteed when the starting distribution is absolutely continuous [Santambrogio, 2015]. Similarly, for the Cartoon dataset, we apply Gaussian noise with a standard deviation of $0.2$ to avoid significantly altering the images.

### E.1    Hyperparameters for CMFGen

Our algorithm, CMFGen, introduces no additional hyperparameters compared to the generative ICNN trained with the solver from Amos [2023], except for the number of LMC steps used when sampling

from $\mathfrak{P}_{w_\theta}$. For a fair comparison, we adopt the same ICNN architecture for both CMFGen and the generative ICNN baseline shown in Figures 13, 5, and 15. We follow the standard ICNN design from Vesseron and Cuturi [2024], which is known to perform well in generative settings. All the 2D experiments in Figure 4 were generated using a common set of hyperparameters, detailed in Table 1. Only the number of particles used (i.e., batch size) varies across experiments. For the 2D distributions Circles, S-curve, Scaled-Rotated S-curve and Diag-Checkerboard we used $1024$ particles, while for Checkerboard, a larger number of particles ($4096$) was used to ensure the stability of the algorithm. The hyperparameters used for the MNIST and Cartoon experiments are given in Table 2. Moreover, we adopt the approach of Amos [2023], where the cost of computing the conjugate at each training step is amortized by initializing the conjugate solver with the predictions of an MLP. This MLP is trained via regression on the outputs of the conjugate solver. The MLP consists of three hidden layers with 128 units each for the 2D experiments and three hidden layers with 256 units each for the MNIST and Cartoon experiment. It is trained using the Adam optimizer with default parameters, and a fixed learning rate of 1e-4 for the 2D experiments and 5e-4 for the MNIST experiment. As explained in Amos [2023], the predictions of the MLP are then refined with a conjugate solver whose hyperparameters are given Table 3 and are kept consistent across all experiments with both CMFGen and the generative ICNN. It is important to note that the number of LMC steps listed in the tables refers to the steps taken to sample from $\mathfrak{P}_{w_\theta}$ starting from uniform noise, as used to generate the figures. During training, however, we reuse the particles sampled from the previous gradient step and apply 200 LMC steps to these particles to form the new batch. The number of LMC steps during training reflects a trade-off between computational efficiency and performance: using significantly fewer steps (e.g., 50) results in degraded performance.

| Hyperparameter | Value |
|---|---|
| $w_\theta$ architecture | dense
$2 \rightarrow 128 \rightarrow 128 \rightarrow 128 \rightarrow 128 \rightarrow 128 \rightarrow 1$
ELU activation functions |
| $w_\theta$ optimizer | Adam
step size $= 0.0001$
$\beta_1 = 0.5$
$\beta_2 = 0.5$ |
| Langevin sampling | number of steps $= 10000$ (=200 during training) |
| number of gradient steps | 1,000,000 |

Table 1: Hyperparameters for CMFGen and the generative ICNN in 2D experiments. The only additional hyperparameter used by CMFGen, compared to the generative ICNN, is denoted in orange.

| Hyperparameter | Value |
|---|---|
| $w_\theta$ architecture | dense
$2 \rightarrow 512 \rightarrow 512 \rightarrow 512 \rightarrow 512 \rightarrow 512 \rightarrow 1$
ELU activation functions |
| $w_\theta$ optimizer | Adam
step size $= 0.0005$
$\beta_1 = 0.5$
$\beta_2 = 0.5$ |
| Langevin sampling | number of steps $= 10000$ (=200 during training) |
| number of gradient steps | 50,000 |
| batch size | 512 |

Table 2: Hyperparameters for CMFGen and the generative ICNN in the MNIST and Cartoon experiments. The only additional hyperparameter used by CMFGen, compared to the generative ICNN, is denoted in orange.

| Hyperparameter | Value |
|---|---|
| Conjugate solver $w_\theta^*$ | Adam
step size with cosine decay schedule
init value = 0.1
alpha = 1e-4
$\beta_1 = 0.9$
$\beta_2 = 0.999$
100 iterations |

Table 3: Hyperparameters for the conjugate solver used in CMFGen and for the generative ICNN (the same hyperparameters are used in all experiments).

## E.2   Hyperparameters for CMFMA

The hyperparameters for our CMFMA algorithm are listed in Tables 4 and 5. We follow standard ICNN architectures, except for the 1D experiments, where we observed discontinuities in the learned potential when using ELU activations. This issue arises because the second derivative of ELU is not continuous, and the CMFMA regression loss involves the Hessian of the network $w_\theta$. To address this, we use Softplus activations with a beta coefficient of 10.

| Hyperparameter | Value |
|---|---|
| $w_\theta$ architecture | dense
$2 \to 128 \to 128 \to 128 \to 128 \to 128 \to 1$
Softplus($\beta = 10.0$) activation functions |
| $w_\theta$ optimizer | Adam
step size = 0.0001
$\beta_1 = 0.5$
$\beta_2 = 0.5$ |
| number of gradient steps | 100,000 |
| batch size | 1024 |

Table 4: Hyperparameters for CMFMA in 1D experiments.

| Hyperparameter | Value |
|---|---|
| $w_\theta$ architecture | dense
$2 \to 128 \to 128 \to 128 \to 128 \to 128 \to 1$
ELU activation functions |
| $w_\theta$ optimizer | Adam
step size = 0.0001
$\beta_1 = 0.5$
$\beta_2 = 0.5$ |
| number of gradient steps | 500,000 |
| batch size | 1024 |

Table 5: Hyperparameters for CMFMA in 2D experiments.

