# OpenReview forum: "Sample and Map from a Single Convex Potential: Generation using Conjugate Moment Measures"
_NeurIPS.cc/2025/Conference — NeurIPS 2025 poster_

### Official Review · Reviewer_Cdjx · 2025-07-01

**Clarity:** 3
**Significance:** 3
**Originality:** 3
**Rating:** 4
**Confidence:** 4

**Summary:**

This paper proposes a new method for generative modeling based on optimal transport and moment measures. The method links the sampling phase and the mapping phase, which are typically treated as completely separate in most generative modeling approaches. First, the authors recall a result on moment measures stating that for any measure ρρ (satisfying certain conditions), there exists a unique convex function $ u $ such that  $ \rho = \nabla u $ # $  e^{−u} $. They then observe, using a simple Gaussian example, that this factorization is not well suited for generative modeling. The authors subsequently propose an alternative factorization: there exists a unique convex function $ w $ such that  $ \rho = \nabla w^* $ # $  e^{−w} $ where $ w^* $ is the convex conjugate of $ w $. They propose learning the convex potential $ w$  using Input-Convex Neural Networks (ICNNs), leveraging recent advances in neural optimal transport solvers, and then sampling from $ e^{-w} $ using a Langevin dynamics scheme. Finally, the authors demonstrate the effectiveness of their method for generative modeling on low-dimensional toy examples and small synthetic image datasets.

**Questions:**

- 1/ If $ \rho = \nabla u$  #  $ \mathfrak{B}_u = \nabla w^* $ #  $ \mathfrak{B}_w $, do we have $ \mathfrak{B}_u^*= \mathfrak{B}_w $?

- 2/ What could be the impact of the fact that your results only hold for absolutely continuous measures, given the manifold hypothesis, which suggests that real data distributions are highly degenerate?

- 3/ Is $ \nabla w^* $ still an OT map?

- 4/ From the experiments, it appears that the results are often noisy. Do you have an intuition to explain this?

**Ethical Concerns:**

["NO or VERY MINOR ethics concerns only"]

**Final Justification:**

I think this is an interesting theoretical paper with a nice new idea. However, the heavy dependence on Input Convex Neural Networks (ICNNs) for the practical realization of the method seems to be a serious limitation. For this reason, I will keep my rating (borderline accept).

**Limitations:**

I think the limitations of the method could be a bit more discussed, relatively to the potential discussion that better situates the contribution already mentioned in section 'Strengths and Weaknesses'.

**Quality:**

4

**Strengths And Weaknesses:**

This is a good theoretical paper that is well written. The maths, as far as I can tell, seem to be correct, and the method appears to be original and novel. Although the experimental results suggest that the method cannot yet compete with state-of-the-art generative modeling approaches—even on relatively simple tasks such as 2D toy examples or MNIST generation—I still consider it a promising new direction with solid theoretical foundations and a strong connection to optimal transport.

That being said, I believe the experimental section could be slightly improved. In my opinion, it lacks commentary on failure cases, such as the checkerboard patterns in 2D, for instance. Additionally, the ICNN architecture used for the MNIST experiment seems overly simple compared to standard models typically applied to this dataset. It would have been helpful to test a more expressive architecture—at least a convolutional neural network (CNN)—to better assess the potential of the method for generative modeling.

Lastly, I think the paper would benefit from a brief discussion that situates the contribution within the broader field of generative modeling. In particular, it would be useful to outline the potential advantages of this method over existing alternatives.

---

> ### Author Rebuttal · Authors · 2025-07-31
>
> Dear Reviewer,
>
> Thank you for taking the time to review our paper. We did our best to answer some of your concerns / questions below.
> >**That being said, I believe the experimental section could be slightly improved. In my opinion, it lacks commentary on failure cases, such as the checkerboard patterns in 2D, for instance.**
>
> We agree that CMFGen does not perform perfectly on the 2D checkerboard dataset and that we could comment more on that. However, note that the generative ICNNs also fail in this experiment (see Figure 13). More generally, the box plots on Figure 13 shows that in low-dimensional settings, our method performs as well as generative ICNNs, and outperforms them on higher-dimensional datasets such as MNIST and Cartoon (see Figures 5, 14, and 15). Since ICNNs are known to be challenging to train [Korotin et al. 2021], we believe the failure on the checkerboard dataset is primarily due to the limitations of the ICNN itself rather than the methodology we propose. We will clarify this briefly in the main paper and provide details in the appendix.
>
>
> >**Additionally, the ICNN architecture used for the MNIST experiment seems overly simple compared to standard models typically applied to this dataset. It would have been helpful to test a more expressive architecture—at least a convolutional neural network (CNN)—to better assess the potential of the method for generative modeling.**
>
> Thank you for the suggestion. We tested convolutional ICNNs, which have been used, for example, in [Korotin et al. 2021] to generate clean images from degraded inputs. However, to our knowledge, they have not been applied in generative modeling settings where the starting distribution is a noise prior. In our experiments, convolutional ICNNs did not outperform plain MLP-based ICNNs. We believe our method could benefit from improved integration of convolutional layers into ICNN architectures.
>
> >**Lastly, I think the paper would benefit from a brief discussion that situates the contribution within the broader field of generative modeling. In particular, it would be useful to outline the potential advantages of this method over existing alternatives.**
>
> At this moment, we do not see the conjugate moment factorization as a direct competitor to current state-of-the-art generative models, as its performance lags behind. The main contribution of our approach lies in linking the sampling process of the prior distribution with the transport of this prior to the data distribution. At the moment we see no other approach than the (conjugate) moments approaches presented in the paper. While this may seem like a conceptual curiosity, we hope the originality of our idea can trigger new ideas in the field.
>
>  Compared to standard generative models like diffusion models, our method offers the following advantages:
> - it does not require arbitrarily choosing a prior distribution
> - it provides a fast access to the generated log-probability, enabling applications such as image completion.
>
> The main limitation is that CMFGen relies on Input Convex Neural Networks (ICNNs), which require further improvements for the method to compete with state-of-the-art generative models.
>
> >**1/ If $ \rho = \nabla u$ # $ \mathfrak{B}_u = \nabla w^* $ # $ \mathfrak{B}_w $, do we have $ \mathfrak{B}_u^*= \mathfrak{B}_w $?**
>
> No, this is not the case. Note that in both formulations, the potential appears both in the transport map—$\nabla u$ for the moment measure factorization and $\nabla w^* $ for the conjugate moment measure—and in the associated Gibbs distribution, namely $\mathfrak{B}_u$ and $\mathfrak{B}_w$ respectively. Therefore, the first equation does not imply the second. For instance, in the Gaussian case (see Propositions 1 and 2), we have $\mathfrak{B}_u^* = \mathcal{N}(0, \Sigma)$, while $\mathfrak{B}_w = \mathcal{N}(0, \Sigma^{1/3})$.
> We did not manage to establish any theoretical relationship between $u$ and $w$, or between $\mathfrak{B}_u$ and $\mathfrak{B}_w$.
>
>
> >**2/ What could be the impact of the fact that your results only hold for absolutely continuous measures, given the manifold hypothesis, which suggests that real data distributions are highly degenerate?**
>
> While the existence is guaranteed for densities, the neural networks must be trained on a stream of i.i.d. samples. This is the same type of result that allows for instance the Brenier theorem to guarantee the existence of an optimal map, and numerical methods to try recovering them. While degeneracies might happen if e.g. the samples do clearly lie on a thin submanifold, as is the case for images if the background is always the same, they can be corrected by adding noise.
>
> For these reasons, we do not consider this assumption to be particularly limiting for practical applications.
>
> >**3/ Is $ \nabla w^*$ still an OT map?**
>
> Yes, this follows from Brenier's theorem: for any convex function $f$ and any probability measure $P$, the gradient map $ \nabla f $ is the optimal transport map from $P$ to the pushforward measure $\nabla f$#$P$ under the squared Euclidean cost. Since $w*$ is convex, it follows that $\nabla w^*$ is indeed an optimal transport map.
>
> >**4/ From the experiments, it appears that the results are often noisy. Do you have an intuition to explain this?**
>
> We added low-variance Gaussian noise to the dataset to ensure absolute continuity of the target probability measures. This explains why the generated samples also appear a bit noisy. Moreover, since our networks do not use convolutional layers, this may further amplify the impression of noise in the generated samples.

---

> > ### Comment · Reviewer_Cdjx · 2025-08-04
> >
> > Thank you for answering my questions. I think this is an interesting theoretical paper with a nice new idea. However, the heavy dependence on Input Convex Neural Networks  (ICNNs) for the practical realization of the method seems to be a serious limitation. For this reason, I will keep my rating.

---

> > > ### Author Response · Authors · 2025-08-04
> > >
> > > Dear Reviewer,
> > >
> > > We are grateful for the time you took to read our rebuttal and for your positive remarks about our work.
> > >
> > > Our method indeed relies heavily on Input Convex Neural Networks (ICNNs), which, although difficult to train, are essential for estimating the convex potential of our factorization in dimensions higher than one. As mentioned in the conclusion of our paper and further discussed with Reviewer CXvw (in their last question), we are considering an approximation of the method to improve scalability,  which we intend to explore in future work.
> > >
> > > We remain available to answer any further concern you might have on our draft.
> > >
> > > Best wishes,
> > >
> > > The authors

---

> > > ### Author Response · Authors · 2025-08-05
> > > **Many thanks for taking the time to read our rebuttal**
> > >
> > > We agree with you that ICNNs remain at the moment difficult to train. We hope however that our emphasis on the synergy when learning noise with transport maps can be useful for the community. Many thanks for having taken the time to read our paper annd rebuttal, and for contributing your review and response, whose comments will be used to clarify our paper further.

---

### Official Review · Reviewer_CXvw · 2025-07-01

**Clarity:** 3
**Significance:** 2
**Originality:** 2
**Rating:** 4
**Confidence:** 4

**Summary:**

This paper introduces conjugate moment measures for sampling. Drawing inspiration from moment measures, which enable the expression of a measure $\rho \in \mathcal P(\mathbb{R}^d)$ as $\rho = \nabla u_\sharp e^{-u}$ for some convex $u: \mathbb{R}^d \to \mathbb{R}$, the paper shows that any measure $\rho \in \mathcal P(\mathbb{R}^d)$ that is absolutely continuous with compact, convex support can also be written $\rho = \nabla w^{\ast}_{\sharp} e^{-w}$, where $w: \mathbb{R}^d \to \mathbb{R}$ is a convex potential and $w^{\ast}$ is the convex conjugate of $w$. The paper additionally shows the existence of such a conjugate moment decomposition for Gaussian measures with non-degenerate covariance matrices and argues, based on examples for two-dimensional Gaussian and one-dimensional Gaussian mixture distributions, that conjugate moment measures are better suited for sampling than moment measures.

The paper then proposes computational strategies for sampling from $\rho$ via its conjugate moment measure decomposition under three different information scenarios: (1) when $w$ is known, (2) when existing samples from $\rho$ are known, and (3) when the unnormalized density of $\rho$ is known. Estimation of $w$ from samples involves a fixed-point algorithm wherein a Brenier potential is computed on each step, either in closed form for 1D distributions or by optimization of an ICNN using neural optimal transport. Estimation of $w$ from a density entails training an ICNN on a collocation loss corresponding to the Monge-Ampere equation (a-la PINNs). In either case, once $w$ is in hand sampling is performed by first using Langevin Monte Carlo (LMC) to sample from $e^{-w}$ and then using $w^*$, identified via gradient-based optimization, to push-forward these samples to $\rho$. The numerical examples in the paper demonstrate proof-of-concept.

**Questions:**

* Regarding the increased computational cost relative to other methods: if we have the density of $\rho$, in order to sample from $\rho$ using a conjugate moment potential we have to (1) train a PINN (which specifically must be an ICNN) to obtain $w$, (2) run LMC to get samples of $e^{-w}$, and then (3) compute $\nabla w^*(x)$ for every sample $x$ generated by LMC. The pipeline for generative modeling with conjugate moment measures is similar. There are many sampling methods that enable the user to generate new samples directly after the training of a neural network (neural ODEs, flow-matching, using PINNs to solve continuity equations, etc.), without the post-processing steps involved in the proposed method. In the density driven-setting, one could run LMC on $\rho$ directly. How does one rationalize the extra costs associated with using conjugate moment measures to sample? Have you compared the quality of samples generated by the conjugate moment measure with those generated by a trained neural network (for dynamic transport) directly or by LMC?

* Theorem 1 guarantees existence of a conjugate moment factorization for absolutely continuous measures $\rho$ with compact, convex support. Many of the distributions in the numerical examples do not satisfy these assumptions. Is this a problem? How do you know, for example, that the fixed-point iteration in CMFGen will converge? Have you tried any examples where it didn't converge?

* It appears that noise is added to the generative modeling training images to guarantee existence of an OT map. Computationally, does the pipeline break down if noise is not added?

* The face image inpainting example seems relatively easy because the mask covers one vertical half of the the image and faces are symmetric about the vertical axis. I could inpaint the images by simply reflecting the unmasked half across the vertical axis. What happens if you mask the top or bottom half of the image instead of a left or right half? Is the quality of the in-painting as good, or does it get degraded?

**Ethical Concerns:**

["NO or VERY MINOR ethics concerns only"]

**Final Justification:**

After discussion with the authors, I agree with them that the primary contribution of this paper is not a new, state-of-the-art method for sampling or generative modeling, but rather a new paradigm in which to approach generative modeling. The conjugate moment potential factorization and the fixed point algorithm they devise to find it are mathematically interesting, and I think that aspects of this work have the potential to inform future generative modeling pipelines. I therefore am increasing my score to a (4), as the numerical results are still somewhat weak and limited.

**Limitations:**

It appears that in the generative modeling setting noise must be added to images before training to ensure the existence of an OT map. This requirement seems like a major limitation for generative modeling, as it would preclude the generation of noiseless images, but it is not discussed outside of one line in the appendix. Do the authors see a way that this step could be avoided? Or do they see a way that their conjugate moment factorization could play an intermediate role in generative modeling pipelines which ultimately produce noiseless images? Further discusson of these limitations and possibilities in the conclusion would help contextualize the results of the paper.

**Paper Formatting Concerns:**

There are places in the manuscript where `citep` and `citet` are not used correctly. E.g., the citations on lines 36-37 should be `citep`'s instead of `citet`'s, and the citation on line 38 should be a `citet`. Please check your citations to ensure correct usage of `citep` and `citet` throughout the manuscript.

**Quality:**

3

**Strengths And Weaknesses:**

## Strengths
* This paper is clearly written and easy-to-follow.

* The result showing existence of conjugate moment measures is interesting and new and follows from a nice fixed-point argument.

* The results and figures concerning the moment potentials and conjugate moment potentials of Gaussians are helpful for building intuition.

* The experimental details are clearly detailed in the appendix and would facilitate reproduction of these results by others.

## Weaknesses
* The argument that moment potentials are unsuitable for practical sampling tasks and that conjugate moment potentials offer a better alternative is primarily based on a relatively narrow set of examples -- Gaussians and Gaussian mixtures in one and two dimensions.  I am not convinced that the behavior in these examples generalizes to non-Gaussian (mixture) and higher-dimensional distributions based on the arguments given in the paper.

* Moreover, moment measures/potentials are not typically used for practical sampling tasks, so from that standpoint the idea that we need an alternative to the moment potential factorization isn't credible.

* The computational effort required to sample via a conjugate moment measure is high compared to other sampling methods, and the extra computation has not been adequately justified.

* The numerical results include no comparisons to widely used or state-of-the art sampling methods. The only comparisons are to ICNNs trained to push forward a Gaussian to $\rho$ directly, whereas state-of-the-art neural network transport approaches usually invoke some kind of dynamics, rather than being "one-shot".

---

> ### Author Rebuttal · Authors · 2025-07-31
>
> We are very grateful for your time spent reading our paper, as well as for your detailed review. We did our best to address your concerns and questions below.
>
> >**The argument that moment potentials are unsuitable for practical sampling tasks and that conjugate moment potentials offer a better alternative is primarily based on a relatively narrow set of examples -- Gaussians and Gaussian mixtures in one and two dimensions. I am not convinced that the behavior in these examples generalizes to non-Gaussian (mixture) and higher-dimensional distributions based on the arguments given in the paper.**
>
> If you were not convinced by our Gaussian experiments, we suggest that you read the argument put forward in L.119-131, which helped us understand better the limitations. In the moment measure factorization, $\nabla u$ must take values, by construction, on the support of the measure 𝜌 to be factorized. If 𝜌 is narrow and concentrated, so must $\nabla u$ be. If $\nabla u$ moves slowly, $u$ looks like a linear potential.  If 𝜌 is narrow and concentrated around 0, then $u$ is almost constant. In both cases, $u$ would be fairly degenerated and extremely spread on its support.
>
> >**Moreover, moment measures/potentials are not typically used for practical sampling tasks, so from that standpoint the idea that we need an alternative to the moment potential factorization isn't credible.**
>
> We agree that moment measures were never highlighted as a means to generate data better.
> Given the importance of generative models in society as a whole, we believed that exploring original ideas to define new generative models was a worthwhile avenue. We looked for models that include, holistically, a parameterized noise description, and naturally found inspiration in moment factorizations (L3-4).
> Moment measures are not widely known within the machine learning community, and one of our contributions is to introduce them to Neurips, and propose an alternative factorization.
>
> >**The numerical results include no comparisons to widely used or state-of-the art sampling methods. The only comparisons are to ICNNs trained to push forward a Gaussian to $\rho$ directly, whereas state-of-the-art neural network transport approaches usually invoke some kind of dynamics, rather than being "one-shot".**
>
> It is true that our algorithm does not match the performance of progressive generative models. However, our approach addresses a different aspect of the problem—namely, finding a suitable prior that facilitates the transport by the transport map. We will clarify.
>
> >**The pipeline for generative modeling with conjugate moment measures is similar. There are many sampling methods that enable the user to generate new samples directly after the training of a neural network (neural ODEs, flow-matching, using PINNs to solve continuity equations, etc.), without the post-processing steps involved in the proposed method.**
>
> Your parallel with Neural ODE is interesting!
>
> In that sense the LMC effort is indeed comparable to the ODE solver effort (parameterized by # steps), with the nuance that LMC uses the gradient of a time-independent *potential*, vs. a time parameterized velocity field for ODE.
>
> For that reason, the convergence of LMC is conceptually simpler than the integration of a vector field with a rotational component.
>
> >**In the density driven-setting, one could run LMC on $\rho$ directly. [...] Have you compared the quality of samples generated by the conjugate moment measure with those generated by a trained neural network (for dynamic transport) directly or by LMC?**
>
> In the case of non–log-concave distributions such as $\rho$, the LMC algorithm lacks theoretical guarantees [Roberts and Tweedie, 1996; Cheng and Bartlett, 2018; Dalalyan and Karagulyan, 2019]. In particular, when the probability density function has multiple local maxima, the generated samples tend to be highly correlated, as particles from the LMC algorithm often become trapped in local basins [Pompe et al. 2020] [Gabrié et al.
> 2022].
>
> In the variational inference framework used in Section 4.3, our focus is on learning the conjugate moment potential, rather than designing a practical algorithm to sample from unnormalized densities. For this reason, we did not compare to other sampling algorithms. To make the method practical, a possible direction would be to combine our approach with MCMC methods to improve the current uniform sampling over the optimization space, in the spirit of the algorithm proposed by [Gabrié et al., 2022].
>
> >**Theorem 1 guarantees existence of a conjugate moment factorization for absolutely continuous measures $\rho$ with compact, convex support. Many of the distributions in the numerical examples do not satisfy these assumptions. Is this a problem?**
>
> While the existence is guaranteed for densities, the neural networks _must_ be trained on a stream of i.i.d. samples. This is the same type of result that allows for instance the Brenier theorem to guarantee the existence of an optimal map, and numerical methods to try recovering them. While degeneracies might happen if e.g. the samples do clearly lie on a thin submanifold, as is the case for images if the background is always the same, they can be corrected by adding noise.
>
> >**How do you know, for example, that the fixed-point iteration in CMFGen will converge? Have you tried any examples where it didn't converge?**
>
> We did not manage to theoretically prove the convergence of the fixed-point iteration in CMFGen. Note that in practice (except in 1D), we run a smoothed version where we do not compute the exact OT map at each iteration but instead take a single gradient step. With this strategy, the algorithm always converged for all the experiments. In 1D, where the OT map is fully estimated at each iteration, we encountered convergence issues in some experiments (distinct from those presented in the paper). However, these were easily resolved using a simple heuristic: averaging the OT maps computed at iterations $n-1$ and $n-2$. This is a failure case that we omitted and will add to the Appendix. It may in fact suggest that using momentum based corrections could be beneficial.
>
> >**It appears that noise is added to the generative modeling training images to guarantee existence of an OT map. Computationally, does the pipeline break down if noise is not added?**
>
> Following your question, we tested our algorithm without adding noise on the MNIST and Cartoon datasets. While the pipeline remained stable and did not break, the visual results were slightly degraded. These cases are interesting because they are not covered by Theorem 1, which assumes absolutely continuous probability measures.
>
> >**The face image inpainting example seems relatively easy because the mask covers one vertical half of the the image and faces are symmetric about the vertical axis. [...] Is the quality of the in-painting as good, or does it get degraded?**
>
> It is not clear whether the experiments mentioned are more challenging for the algorithm. We conducted the two experiments you suggested—masking the top half and masking the bottom half of the image—and in both cases, the algorithm consistently filled in the masked regions. The results were comparable in quality to those obtained with horizontal masking, as shown in Figure 17. When masking the bottom part of the image, the algorithm tends to produce samples that closely match the original. However, when masking the top part, features such as hair color can become ambiguous—especially in cases where the cartoon character has very short hair—leading the algorithm to generate cartoons with different hair colors or hairstyles. Unfortunately we cannot share these images due to the rebuttal process in place this year.
>
> >**Do the authors see a way that this step could be avoided? Or do they see a way that their conjugate moment factorization could play an intermediate role in generative modeling pipelines which ultimately produce noiseless images?**
>
> We currently do not see a way to remove the added noise without degrading performance, but this is intrisically due to the degeneracy (uniform background) of the data. A possible direction to improve scalability would be to use our method to estimate the conjugate moment potential of the distribution $\rho$, and then retain only the corresponding Gibbs distribution as a learned log-concave prior. Flow Matching combined with mini batch Optimal Transport ([Tong et al. 2023], [Pooladian et al. 2023]) could then be used to learn a generative model between this prior and the target distribution. This approach would approximate our current method and may offer better scalability. This however would involve two neural networks (one for learning the prior, one for transport), and thus move slightly away from the original theory of conjugate moment measure factorization, which relies on a single convex potential.

---

> > ### Comment · Reviewer_CXvw · 2025-08-05
> >
> > I thank the authors for their detailed and thorough responses. Based on their response and their discussion with other reviewers, I see more clearly the merit of the paper's contributions, so I will increase my score.

---

### Official Review · Reviewer_dw2D · 2025-07-02

**Clarity:** 3
**Significance:** 3
**Originality:** 4
**Rating:** 4
**Confidence:** 4

**Summary:**

The standard approach in generative modeling separates noise sampling (e.g., Gaussian) from mapping (e.g., flows). The authors propose an alternative that links sampling and mapping, inspired by moment measures, which connect log-concave distributions and convex potentials. However, this method is impractical for common tasks. Instead,  ρ is factorized as ∇w* ♯ e^−w, with w* the convex conjugate of w.  This is called the conjugate moment measure approach. Using optimal transport and input-convex neural networks, w is recovered from samples, even when ρ’s density is only known up to a constant. This yields more intuitive results in practice.

**Questions:**

I have the following questions:

(1) Is the conjugate moment measure factorization given in Theorem 1 unique?

(2) What is the  computational complexity the proposed algorithm?

(3) How scalable the algorithm is?

(4)  I wonder if the authors can comment on the statistical errors of the generated distribution
      based on the proposed method.

**Ethical Concerns:**

["NO or VERY MINOR ethics concerns only"]

**Limitations:**

Yes.

**Paper Formatting Concerns:**

This is a technically dense paper. It would be helpful if the aiuthors could provide a high-level and less technical description of the main idea of the proposed method.

**Quality:**

3

**Strengths And Weaknesses:**

Stengthes: This paper presents a novel approach to generative modeling by proposing an alternative to the traditional two-step process of noise sampling and mapping. Drawing inspiration from the theory of moment measures, the authors explore a framework that directly couples the sampling and mapping steps.  Specifically,  the paper introduces the idea of "conjugate moment measures," where the target distribution is represented via the Monge map associated with the convex conjugate of a potential function. The authors use optimal transport solvers and parameterize the potential using input-convex neural networks.

Weakness: The main weakness of this work is that the computational cost is not dsicussed in detail. For sampling
sampling from ρ using its Conjugate Moment Factorization, it requires concave maximization problem:∇w∗(y) = arg supx ⟨x, y⟩ − w(x) in each iteration. It is not how expensive this is computaitonally, especially in high-dimensional settings. Second, there is no discussion on the advantages/disadvantages of the proposed method relative to the existing generative methods, such as diffuion and continuous normalizing flows. Finally, the experimental results are not convicining. While the results show some promise of the method, more experiments with higher-dimensional image data will  significantly strengthen the paper.

---

> ### Author Rebuttal · Authors · 2025-07-31
>
> Many thanks for taking the time to review our paper and for your valuable feedback.
>
> >**The main weakness of this work is that the computational cost is not dsicussed in detail. For sampling sampling from ρ using its Conjugate Moment Factorization, it requires concave maximization problem:∇w∗(y) = arg supx ⟨x, y⟩ − w(x) in each iteration. It is not how expensive this is computaitonally, especially in high-dimensional settings.**
>
> We agree that this aspect is not discussed in the paper and we will add a paragraph in the revised version.
>
> As you noted, the problem is a well-posed concave maximization that must be solved at inference time but also during training at each gradient step — not by differentiating through the solver, but by using the optimal solution to get the gradient.
>
> This challenge is inherent to neural OT solvers ([Makkuva 2020],  [Amos 2023]). The recent solver we use from [Amos 2023]  amortizes the cost of computing $\nabla w^\*$
> by training an MLP that, given $y$, predicts $\nabla w^\*(y)$ **L.687** in order to initialize the solver. This prediction is then refined by running 30 optimization steps (as proposed in [Amos 2023]) to improve accuracy. This refined value is used both in the loss (step 5 of Algorithm 1), applying `jax.lax.stop_gradient` to treat it as constant, and to train the MLP for better approximations. At inference, we can again amortize the cost by using the MLP prediction refined with these optimization steps to get a fast and accurate approximation of $\nabla w^*$.
>
>
> >**Second, there is no discussion on the advantages/disadvantages of the proposed method relative to the existing generative methods, such as diffuion and continuous normalizing flows.**
>
> At this moment, we do not see the conjugate moment factorization as an immediate competitor in the field of generative modeling, and do not wish to present it that way, for the obvious reason that our performance is lagging behind.
> While we believe this is very likely due to the limitations arising when training ICNNs. Our paper simply asks whether one can propose a generative model that is able to include **both** noise distribution **and** transport in a single parameterization. At the moment we see no other approach than the conjugate moments approaches presented in the paper. While this might be a curiosity, we hope the originality of our idea can trigger new ideas in the field.
>
> >**Finally, the experimental results are not convicining. While the results show some promise of the method, more experiments with higher-dimensional image data will significantly strengthen the paper.**
>
> We agree that the current method does not yet compete with state-of-the-art generative models. However, relying solely on ICNNs, it improves upon results generated by generative ICNNs. See Figure 5 for comparisons on MNIST, and Figures 14 and 15 for results on the Cartoon dataset. For a discussion on the limited expressiveness of ICNNs in generative modeling, see [Korotin et al., 2021].
>
> >**(1) Is the conjugate moment measure factorization given in Theorem 1 unique?**
>
> We did not manage to answer this question in the general case. However, for the Gaussian case, we proved the existence of a unique fixed point within the class of quadratic functions (whose Gibbs distributions are Gaussian). This is a challenging theoretical question that we are currently investigating.
>
> >**(2) What is the computational complexity the proposed algorithm?**
>
> For CMFGen, the algorithm complexity per gradient step is `O(d PB(T_{LMC}+T_{conj}))` with `B` the batch size, `P` the number of parameters, `T_{LMC}` the number of steps used for `LMC`, and `T_{conj}` the number of steps used for computing the convex conjugate `d` dimension. For CMFMA the complexity is `O(d PB(T_{LMC}+T_{conj}+P))` due to hessian computations.  For comparison, a supervised training of a NN is typically `O(PB)`.
>
> >**(3) How scalable the algorithm is?**
>
> This question can be interpreted in multiple ways.
>
> Perhaps the most obvious bottleneck in our implementation lies in data dimensionality, for two reasons:
> - $w_{\theta}$ is parameterized as an ICNN. In high $d$, ICNNs are known to be hard to train. While they encode desirable convexity properties, the constraints in their parameters prevent established optimizers (e.g. ADAM) from converging quickly. While we do obtain reasonable results (our paper is, to the best of our knowledge, the first to train ICNNs in the pixel space of MNIST) there is still a gap w.r.t non-constrained models.
> - Our approach requires evaluating conjugate functions at all points in a mini-batch. While the function itself is convex and the optimization problem is therefore well posed, increase in dimension will typically mean that more effort is needed to evaluate these conjugate functions $w_{\theta}^\*$.
>
>
> >**(4) I wonder if the authors can comment on the statistical errors of the generated distribution based on the proposed method.**
>
> Practically speaking, the box plots on the Sinkhorn divergence (Figure 13) demonstrate that the generated distribution was close to the target, indicating good empirical performance.
> Theoretically, the statistical error depends on how well the method approximates the conjugate moment potential, which in turn hinges on whether the fixed point is found—something we have not yet proven. In theory, at least, the use of ICNNs is not a limitation here, as they are universal approximators of convex functions.

---

> > ### Comment · Reviewer_dw2D · 2025-08-09
> >
> > I appreciate the responses by the authors, which have largely addressed the main issues I raised.
> > I will increase my score by 1 point.

---

### Official Review · Reviewer_CFgx · 2025-07-03

**Clarity:** 3
**Significance:** 4
**Originality:** 4
**Rating:** 5
**Confidence:** 3

**Summary:**

The paper proposes to a novel approach to generative modeling, using a variant of optimal transport. Rather than relying on a heuristically chosen initial distribution, which is possibly the biggest hyperparameter in generative modelling, they propose a method which constructs the base distribution along with the pushforward map to the data distribution. They build upon known theoretical tools like the brenier potential and moment measure to eliminate the limitations of typical optimal transport based methods which are prone to create very ill-conditioned maps. They demonstrate the results on some synthetic low dimensional datasets as well as MNIST and Cartoon dataset.

**Questions:**

High level questions:
1. Can you please provide a complete and formal statement of Breiner's thorem that is being used either in the main paper or in Appendix?
2. From what I understand, although the theory holds for convex potentials, none of the methods are actually using convex potentials. Is that correct? If not, how are you ensuring that $w_\theta$ are convex? If yes, how is convex conjugate / legendre transform interpreted for a non-convex function?
3. Would it be possible to compare performance with other energy-based modelling methods (like say score matching) in a synthetic setting for example over an exponential family?
4. Line 214: How is $w_\theta$ initialized to be roughly $\lVert \cdot \rVert^2$? It does not feel necessary to have this initialization for the proposed method, can you explain choice of this initialization?

Questions/comments about the proofs:
1. Line 404: $w$ is convex only if $\Omega$ is convex. Is this assumed?
2. Line 405: How is $\nabla w$ defined? $w$ seems to be discontinuous on boundary of $\Omega$.

**Ethical Concerns:**

["NO or VERY MINOR ethics concerns only"]

**Final Justification:**

I would like to thank the authors for response to my concerns, all the ones which can be answered without increasing the scope of the paper drastically and unrealistically have been answered. I would like to keep my original score and would urge the AC to lean towards acceptance if the paper is right on the border.

**Limitations:**

yes

**Paper Formatting Concerns:**

1. Lines 69-74: $f^*$ notation is overloaded to mean conjugate as well as optimal point.
2. Line 85: (Gibbs) log concave probability measure -> log-concave (Gibbs) probability measure
3. Lines 91: Does not supported on hyperplane mean that support is not contained in any hyperplane?
4. Lines 89-92: Isn't boundary a measure zero set, implying that if $u$ is only discontinuous at the boundary then it is still essentially continuous? I do not see how this is relevant to rest of the paper.
5. Lines 178-179: From what I understand LMC only ever provides an approximation, these lines seem to suggest that it matches exactly to the final distribution after a warm-up period.
7. Line 428: Is $\mathscr{C}(\Omega)$ the set of convex functions on $\Omega$? $\mathcal{C}(\Omega)$ is used for same purpose on line 71.
6. Formatting seems off, especially some line numbers are missing etc, for example line 425-426, 428-429.

**Quality:**

3

**Strengths And Weaknesses:**

The paper developes a critical idea to eliminate one of the more important hyper parameters in generative modelling, the initial distribution. The known approaches to picking this distribution are mainly heuristic, and this proposed alternative provides a good approach to this problem. The main result of the paper is the existence of conjugate moment potential, which is proved via a fixed point theorem applied to the Brenier potential map. I found the proof to be fairly involved.

Paper presents a fairly convincing argument about why conjugate moment potentials are preferred over moment potentials. The argument is limited to a Gaussian distribution, but suffice to convince the ill-conditioned nature of moment potentials.

The paper has limited experimental coverage with focus on low-dimensional synthetic datasets along with some experiments on high dimensional datasets like MNIST and Cartoon. It would be interesting to see how the experiments scale, but I think current experiments suffice for the purpose of this paper whose main goal is to explore a novel modelling idea.

---

> ### Author Rebuttal · Authors · 2025-07-31
>
> Many thanks you for your encouraging comments. We appreciate that you have recognized our contribution, which proposes to reassess the link between noise distribution and transport.
>
> >**Can you please provide a complete and formal statement of Brenier's theorem that is being used either in the main paper or in Appendix?**
>
> We will include a formal description of the Brenier theorem in our appendix.
>
> >**From what I understand, although the theory holds for convex potentials, none of the methods are actually using convex potentials. Is that correct? If not, how are you ensuring that $w_\theta$ are convex? If yes, how is convex conjugate / legendre transform interpreted for a non-convex function?**
>
> Thank you for the question. In our case, the potentials $w_\theta$ are indeed convex with respect to their input by construction, as we rely on Input Convex Neural Networks (ICNNs) [Amos et all. 2017], this is mentioned L.170. We will add a more detailed explanation of the ICNN construction in the appendix for clarity. Regarding the convex conjugate: since our potentials are convex by design, the Legendre transform is well-defined and its gradient can be reliably computed by maximizing the concave problem defined L.181 using a solver. We use an off-the-shelf L-BFGS implementation (L.182).
>
> >**Would it be possible to compare performance with other energy-based modelling methods (like say score matching) in a synthetic setting for example over an exponential family?**
>
> Our method does not currently outperform state-of-the-art generative models, and we expect that advanced score-matching-based models will achieve better performance than ours. We believe this is primarily due to our reliance on ICNNs, which, while advantageous for parameterizing convex functions, are known to be challenging to train ([Korotin et al. 2021]). For this reason, we compare our method to other generative ICNNs using the Sinkhorn divergence as a metric (see boxplots in Figure 13). Make the method scalable will require improving further the training of ICNN architectures, an important research topic.
>
> >**Line 214: How is $w_\theta$ initialized to be roughly $\lVert \cdot \rVert^2$? It does not feel necessary to have this initialization for the proposed method, can you explain choice of this initialization?**
>
> This initialization was proposed in [Bunne et al. 2022] and is akin to initializing with the identity map, but you are correct, one can consider other more data informed initializations, and this topic has been the subject of great research in [Hoedt, Klambauer, 23] but this is not yet useful for OT maps.
> [ Hoedt, Klambauer, Principled Weight Initialisation for Input-Convex Neural Networks, Neurips 23]
>
> >**Line 404: $w$ is convex only if $\Omega$ is convex. Is this assumed?**
>
> Yes, $\Omega$ is assumed to be convex, as stated in Theorem 1. We will also clarify this in the appendix when presenting the proof.
>
> >**Line 405: How is $\nabla w$ defined? $w$ seems to be discontinuous on boundary of $\Omega$.**
>
> The function $w$ is differentiable in the interior of $\Omega$, so taking its gradient there is well-defined. On the boundary, $w$ is discontinuous by definition, as it takes infinite values outside $\Omega$, and thus its gradient is not defined there. However, note that the boundary of the bounded convex set $\Omega$ has Lebesgue measure zero, so this does not affect the proof.
>
> >**Formatting issues**
>
> Thanks for spotting these typos — we will take them into account in the revised version.
>
> >**Lines 91: Does not supported on hyperplane mean that support is not contained in any hyperplane?**
>
> It means that the support does not **strictly** include any hyperplane (or, more generally, does not put positive mass on a subset of lower dimension than the ambient space, or, more loosely, that it has a density w.r.t. the Lebesgue measure) This is a classic assumption to obtain results in OT.
>
> >**Isn't boundary a measure zero set, implying that if $u$ is only discontinuous at the boundary then it is still essentially continuous? I do not see how this is relevant to rest of the paper.**
>
> The boundary of the bounded, convex set $\Omega$ is a null set for the $n$-dimensional Hausdorff measure, but the definition of essential continuity in this context requires the discontinuity set to have zero $(n-1)$-dimensional Hausdorff measure. Since the boundary has positive $(n-1)$-dimensional measure, $u$ is not essentially continuous. This result is not directly relevant to our contributions; it was intended to introduce [Cordero-Erausquin and Klartag, 2015], and the problem they studied, as they made a significant contribution to the moment measure factorization with this result.
>
> >**From what I understand LMC only ever provides an approximation, these lines seem to suggest that it matches exactly to the final distribution after a warm-up period.**
>
> You are right, the LMC algorithm is the Euler discretisation of the Langevin diffusion process and this discretization induces a bias that can be controlled for smooth log concave distributions [Dalalyan 2017]. We will add a note on that bias in the revised version.
>
> >**Line 428: Is $\mathscr{C}(\Omega)$ the set of convex functions on $\Omega$? $\mathcal{C}(\Omega)$ is used for same purpose on line 71**
>
> It is actually the Banach space of continuous functions over a compact set, so no convexity assumption is involved. While we used a different font, we will introduce explicit notation such as \(\operatorname{cvx}(\Omega)\) to clarify these distinctions.

---

> > ### Comment · Reviewer_CFgx · 2025-08-07
> > **Response to Authors**
> >
> > I would like to thank the authors for detailed response, which answers almost all of my concerns. Just two comments -
> >
> > I was suggesting exponentially families as a way to avoid dependency on ICNN in a toy setting. If that does not work, it might be worth looking at a toy setting where you can isolate conjugate moments vs classical generative modelling methods to get a more ablated comparison.
> > The point about LMC was not just discretization bias but also about convergence (LMC only converges as $t \rightarrow \infty$ )

---

> > > ### Author Response · Authors · 2025-08-08
> > > **We understand better your point.**
> > >
> > > Many thanks for your time reviewing our paper and our response.
> > >
> > > Your point on EBM is an interesting one, which we had not considered. In essence one can see flow models, conjugate moment measures and EBM as three different approaches on how to generate.
> > > - FM start from arbitrary noise and push it through one (in the case of OT maps, eg using ICNN) or multiple (ODE paramaterization of velocity) steps; the noise choice (typically Gaussian) is crucial since the map is fitted with it.
> > > - CMM (our work) merges noise and push-forward in one single convex potential;
> > > - EBM do not really emphasize noise (it might be needed to sample, ultimately, but no assumption is required on it), but model instead directly the target density.
> > >
> > > While we focused on links with OT models with ICNN, we may indeed consider an EBM baseline on the other end of this scale to report performance differences. While we no longer have the time to do this now, we will push this for comparison in the final draft on experiments with e.g MNiST. We will also add the discussion above.
> > >
> > > Many thanks for staying available during this rebuttal and for your numerous comments that we commit to incorporating in the final draft.

---

### Official Review · Reviewer_EwEU · 2025-07-08

**Clarity:** 4
**Significance:** 2
**Originality:** 4
**Rating:** 5
**Confidence:** 4

**Summary:**

This paper develops a new, mathematically-motivated method for generative modeling. The approach is based on a so-called moment measure factorization of Cordero-Erausquin and Klartag [2015] that provides a factorization of any measure $\rho$ as $\rho = \nabla u \sharp e^{-u}$ where $u$ is a convex potential function. The authors observe that this factorization can be ill-conditioned, and instead propose to leverage a novel, related factorization $\rho = \nabla \omega^* \sharp e^{-\omega}$ where $\omega$ is convex and $\omega^*$ denotes its convex conjugate. The approach is demonstrated on several simple numerical examples, and some theory is proven demonstrating the validity of the approach.

**Questions:**

1. In the introduction, the authors comment that $\omega^*$ is the Brenier potential linking $e^{-\omega}$ to $\rho$. Can the authors describe, then, why the method is different from optimal transport? Is it simply because we also learn the source distribution? We know that the optimal transport map can be hard to learn, and so many generative modeling approaches are intentionally based on sub-optimal transport to simplify the estimation problem. Can the authors explain clearly why it is a "good idea" to try to estimate the optimal transport map if the goal is "only" generative modeling?

2. Can the authors explain the computational efficiency of their approach? The iterative optimal transport computation in Sec. 4.2 seems very expensive.

3. Can the authors comment on the feasibility of their sampling method from Sec. 4.3? The sampling method requires Langevin samples from the target measure $\rho$ to learn. This seems to be a chicken and an egg problem, because the method's goal is to generate samples from $\rho$. From this perspective, I'm having trouble seeing if the method will really work in practice. Is there a way to make it iterative and sample from the current estimate of $\rho$, using the fact that $\omega$ is convex? Will this suffer from mode collapse?

4. Can the authors get their method to work better in the numerical examples? I'm a firm believer that we shouldn't require SoTA image results in every paper on generative modeling, but I do think if the method doesn't really work on the 2d checker, it indicates a problem. For publication, I think the authors need to have compelling results on some simple examples. They should also try to quantify their results in some way, such as FID on images and metrics such as KL or Sinkhorn in low-d. The authors should also tone down the claims in the text that the method works well, which are sort of contradicted by their figures.

**Ethical Concerns:**

["NO or VERY MINOR ethics concerns only"]

**Final Justification:**

After reading all reviews and rebuttals, I have been convinced that the conceptual innovation is strong enough to outweigh the poor performance on the numerical examples.

**Limitations:**

Yes.

**Paper Formatting Concerns:**

None.

**Quality:**

3

**Strengths And Weaknesses:**

**Strengths**
- The paper is very well-written and is a pleasure to read.
- The idea is novel and interesting, and leverages deep mathematical results that have not yet been applied in the generative modeling space, implying a significant level of originality.
- The method is mathematically very well-motivated and grounded, which is refreshing, particularly in a space where engineering advancements can overpower theoretical ones.


**Weaknesses**
- The notation can at times be difficult to read -- for example $*$ is used both to denote the optimizer and the convex conjugate, and there are several usages of related letters such as $\mathcal{B}$ and $\mathfrak{B}$ for distinct concepts.
- The method seems to be very expensive numerically, particularly given its relation to optimal transport. Moreover, the sampling variant requires the ability to sample from the target, which feels circular.
- Most significantly, despite its elegance, the method doesn't work very well: even on the two-dimensional checker example, for which we can now obtain very good results using flow matching or diffusion models in a matter of minutes, the results are pretty poor. The results on higher-dimensional datasets like images are unfortunately quite bad, even for simple cases such as MNIST.
- Very minor comment, but the authors miss a few of the major initial references on flow matching, namely https://arxiv.org/abs/2209.15571, https://arxiv.org/abs/2303.08797, and https://arxiv.org/abs/2209.03003

---

> ### Author Rebuttal · Authors · 2025-07-30
>
> Many thanks you for your time and for your kind comments on the writing, novelty, and theoretical grounding of our work. We address your questions and concerns below.
>
>
> > **The notation can at times be difficult to read -- for example $*$ is used both to denote the optimizer and the convex conjugate, and there are several usages of related letters such as $\mathcal{B}$ and $\mathfrak{B}$ for distinct concepts.**
>
> We agree that the $\mathcal{B}$ - \mathcal{B} notation for the Brenier potential and $\mathfrak{P}$ - \mathfrak{P} for Gibbs distribution look similar. Following your comment, we have changed the notation of the Gibbs potential to $\mathfrak{G}$ - \mathfrak{G}.
>
> Regarding the star notation, we make a distinction between the optimizer denoted $\star$ and the convex-conjugate operator, denoted *. While we agree that the symbols look similar (less so in the paper than in markdown), these are standard notations, and we prefer to keep them as they are.
>
> >**The results on higher-dimensional datasets like images are unfortunately quite bad, even for simple cases such as MNIST**
>
> We acknowledge that our results are far below from the state-of-the-art in generative modeling.
> However, compared to the existing OT approaches that use ICNNs to generate data from Gaussian (e.g. [Amos’23], [Makkuva et al. 2020], [Korotin et al. 2020]) our method improves the results on the 2D datasets, MNIST and Cartoon, while also relying exclusively only on ICNNs. The improvement is particularly visible on MNIST (see Figure 5 for comparison).
>
> >**Very minor comment, but the authors miss a few of the major initial references on flow matching**
>
> Thank you for these additional references, we have added them. Please feel free to suggest any others you believe could be relevant.
>
> >**1.  In the introduction, the authors comment that $\omega^*$ is the Brenier potential linking $e^{-\omega}$ to $\rho$. Can the authors describe, then, why the method is different from optimal transport? Is it simply because we also learn the source distribution?**
>
> Yes, exactly, in the classical optimal transport (OT) problem the source and target distribution are fixed and the objective is to estimate the OT map between these distributions (gradient of a convex potential). In the (conjugate) moment factorizations, **both source distribution and transport map** are uniquely parameterized by the same convex potential (L.37, L.206)
>
> >**We know that the optimal transport map can be hard to learn, and so many generative modeling approaches are intentionally based on sub-optimal transport to simplify the estimation problem. Can the authors explain clearly why it is a "good idea" to try to estimate the optimal transport map if the goal is "only" generative modeling?**
>
> This is a great point. In this paper, our hands are tied with the “ideal” OT parameterizations of transport, by a convex potential, because this is what the factorization theorem dictates.
>
> While having a convex potential helps with LMC sampling, it is indeed a burden when computing the map. We leave the door open on using our method as a first pre-processing step, in order to learn first a “good” prior, before using a more advanced generative tool.
>
> >**2. Can the authors explain the computational efficiency of their approach? The iterative optimal transport computation in Sec. 4.2 seems very expensive.**
>
> The proposed method inherits the efficiency (and limitations...) of neural OT solvers.
>
> Algorithm 1 mirrors the standard procedure for estimating the OT map between two distributions [Amos’23]. The only difference is that, in our case, the (noise) source distribution is re-updated at each iteration. Specifically, in line 4 of Algorithm 1, instead of drawing a minibatch from a fixed distribution, we sample from the evolving distribution using the Langevin Monte Carlo (LMC) algorithm. Re-ampling from this moving, log-concave distribution is the only overhead compared to a standard OT solver.
> To mitigate this cost, we speed up LMC by reusing particles from the previous iteration as a warm start (L.219 in the paper). Furthermore, to compute $\tilde{x}$ at line 5, we leverage the latest OT solver by  [Amos’23], which warm-starts the conjugate solver using predictions from a trained MLP, significantly accelerating the overall algorithm, as noted in Appendix (L.693).
>
> > **3.  Can the authors comment on the feasibility of their sampling method from Sec. 4.3? The sampling method requires Langevin samples from the target measure $\rho$ to learn. This seems to be a chicken and an egg problem, because the method's goal is to generate samples from $\rho$. From this perspective, I'm having trouble seeing if the method will really work in practice. Is there a way to make it iterative and sample from the current estimate of $\rho$, using the fact that $\omega$ is convex? Will this suffer from mode collapse?**
>
> In the variational inference framework used in Section 4.3, we do not require samples from $\rho$. Instead, we assume access to the unnormalized densities associated to $\rho$ i.e. for a given sample $x$, we can require access to $q(x)$ where $q(x) \propto \rho(x)$. The algorithm does not train on samples from $\rho$ but rather on couples $(x, q(x))$ where $x$ is sampled in a more or less clever way. In our experiments, $x$ is sampled uniformly over the domain, and Section 4.3 focuses on learning the conjugate moment potential in this context. To make the method practical,  an idea would be to combine this algorithm with MCMC methods to draw samples $x$ smartly in the same spirit as the methods explored in [Gabrié et al. 2022]. We will include these clarifications in the revised version.
>
> >**4. Can the authors get their method to work better in the numerical examples? I'm a firm believer that we shouldn't require SoTA image results in every paper on generative modeling, but I do think if the method doesn't really work on the 2d checker, it indicates a problem. For publication, I think the authors need to have compelling results on some simple examples.**
>
> We agree that CMFGen does not perform perfectly on the 2D checkerboard dataset. However, note that the generative ICNNs also fail in this case (see Figure 13). More generally, the box plots on Figure 13 shows that in low-dimensional settings, our method performs as well as generative ICNNs, and outperforms them on higher-dimensional datasets such as MNIST and Cartoon (see Figures 5, 14, and 15). Since ICNNs are known to be challenging to train [Korotin et al. 2021], we believe the failure on the checkerboard dataset is primarily due to the limitations of the ICNN itself rather than the methodology we propose. In response to your comment, we experimented with larger ICNN architectures—specifically, models with 6 hidden layers of size 128 and 5 hidden layers of size 512—to improve performance. For now the gains were not significant, but we plan to continue exploring this.
>
> >**They should also try to quantify their results in some way, such as FID on images and metrics such as KL or Sinkhorn in low-d. The authors should also tone down the claims in the text that the method works well, which are sort of contradicted by their figures.**
>
> Note that we use the Sinkhorn divergence to compare our method to generative ICNNs (see Figure 13), where boxplots provide a visual summary of the results. From our perspective, the method performs well—this is, to our knowledge, the first time MNIST samples have been successfully generated using ICNNs (see Figure 5 for a comparison between our results and samples generated by a generative ICNN). That said, we acknowledge that, when compared to state-of-the-art generative methods, the results are not impressive. We will make this distinction clearer in the revised version.

---

> > ### Comment · Reviewer_EwEU · 2025-08-05
> >
> > I thank the authors for their detailed rebuttal. I am intrigued by the method and believe it to be a very nice mathematical contribution -- even moreso after reading the replies to the other reviewers. I do remain concerned about the empirical performance of the approach on such simple datasets as the checkerboard and MNIST. I understand the author's point that training ICNNs is hard, and that their performance is limited by a need to do so. Nevertheless, estimating convex potentials is fundamental to their approach, and I see this as a serious limitation. For example, one reason the community has moved on from discrete normalizing flows is because of the need to use invertible network parameterizations that are much less expressive than the unrestricted forms that can be used with flow and diffusion-based models. As it stands, the numerical experiments essentially demonstrate that the method doesn't really work on the problems it is designed to solve.
> >
> > Despite this criticism, I am inclined to recommend the paper for acceptance, given its clarity and conceptual innovation.
> >
> > One more minor comment -- I find all of \mathfrak quite challenging to read, but I will leave it up to the authors if they prefer to use it.

---

> ### Author Response · Authors · 2025-08-06
> **Many thanks for taking the time to read our response**
>
> We are grateful for your support and constructive criticism.
>
> We agree that our method is not immediately competitive, but we hope that the originality of our mathematical result and proof of concept implementation can also encourage others to reconsider and explore alternatives to the Gaussian + ODE/SDE flow paradigm that has come to completely dominate generation. In particular, we genuinely believe that the choice of the sampling distribution is an understudied area, and we believe our idea is quite innovative in that sense.
>
> Since you mention normalizing flows, we also noticed recently a resurgence of such approaches, see e.g
>
> arxiv/2412.06329 Normalizing Flows are Capable Generative Models
>
> We will carefully incorporate all of the points we have discussed in this rebuttal to clarify the message of our draft. Many thanks for your time and consideration, and for helping us improve our paper.
>
> thanks!
>
> the authors
>
> PS: we will explore a simpler notation than mathfrak for the Gibbs distribution.

---

### Note · Authors · 2025-08-12

Dear AC, SAC and Reviewers,

We are happy to take this opportunity to thank again all reviewers for their interest in our work. We are grateful for their time looking at our paper during these 2 months.

While we have no specific final remarks, we would like to thank all 5 reviewers for their highly valuable, actionable feedback. We have used their comments during the discussion to improve our draft.

We are thankful that all our answers were well received, as evidenced by the fact that Reviewers **CXvw** and **dw2D** mentioned increasing their score while Reviewers **EwEU** **CFgx** **Cdjx** reiterated their appreciation and encouraging comments.

_(minor note: Reviewer **dw2D** mentioned `"I will increase my score by 1 point"` but this has not been registered in the system, at the moment, on our side)_

During this rebuttal phase, we have made a few changes to the draft to answer their remarks:
- Clarified (**EwEU, dw2D, Cdjx**) that we rely on ICNN baselines to keep message coherent as a comparison with other OT methods, mentioning in particular that the MNIST results we propose (Fig. 5) are to our knowledge the first time ICNNs carry that task reasonably well.
- Clarified (**CFgx, Cdjx, CXvw**) that neither the original moment factorization, nor the conjugate moment factorization we propose should be seen as an immediate competitor to flow models, but instead as the first approaches that jointly parameterizes both the noise distribution and the transport in a single ICNN potential.
- Clarified (**CXvw**) why, intuitively, the moment measure factorization results in a flat Gibbs factor when the source is peaked (and inversely), making the simple argument that our conjugate moment approach is more robust, as Gibbs factor and source would have similar support.
- Included discussion on how flow / (conjugate) moment measures / EBM can be positioned (as discussed with Rev. **CFgx**) as approaches that make the most to the least assumptions on noise distribution.
- Tested our algorithm on two other inpainting tasks using top-half and bottom-half masks in response to Reviewer **CXvw**. We can report that in both cases, our algorithm consistently filled in the masked regions with quality comparable to horizontal masking (Fig. 17).

We have also taken advantage of the reviewers' feedback to correct our draft in many more places, and we are grateful for their careful reading.

The authors

---

### Decision · Program_Chairs · 2025-09-17

**Decision:**

Accept (poster)

**Comment:**

This work studied two factorizations of a density $\rho$ through a convex function $u$ or $w$:
- moment measure due to Cordero-Erausquin and Klartag [2015]: $\rho = (\nabla u)_{\sharp} e^{-u}$
- conjugate moment measure (proposed): $\rho = (\nabla w^*)_{\sharp} e^{-w}$

Existence of the conjugate moment measure is established under reasonable assumptions and a fixed-point algorithm is proposed to estimate $w$, when the density $\rho$ is available either in form or in samples or in score. Numerical experiments were conducted to show the feasibility of the proposed factorization as a generative modelling tool.

The reviewers all praised the originality of the idea, the presentation and clarity of the paper and its potential to inspire new developments down the road. The reviewers also expressed concerns on the sub-par empirical performance when applied to generative modelling. Nevertheless, the consensus is that the originality of the idea outweighs the limited empirical results, which I concur. In the end all reviewers recommended (weak) acceptance, and we believe it'll serve the community much good by making the ideas  in this work more widely known.

Some extra comments for the authors to consider incorporating:

- The argument against moment measures in terms of the support is somewhat weak and hypothetical. What is wrong when $u$ is spread out or near constant? The authors also did not exclude the possibility of other examples that would demonstrate similar issues for the conjugate moment measure (at least from the same arguments in Lines 119-131 it follows that $w$ would have concentrated support, which is not necessarily easy to capture through ICNNs).

- Is generative modelling really the right application for the two factorizations? Learning the optimal transport map with two fixed densities is already non-trivial, let alone learning the OT map together with a tied source distribution?

- Related to above, the following issue is somewhat neglected: the conjugate moment measure approach is more expensive during inference time, since we only learn $w$ while $\nabla w^*$ is required for each generation. In this respect, the moment measure approach seems to be more economical. Perhaps the authors should explicitly discuss this tradeoff.

- The argument against a fixed Gaussian source distribution is weak. There is nothing wrong by itself to fix the source distribution to be Gaussian, unless one can compellingly show otherwise (not the case from the authors experiments). In this regard, the following work that learns a source distribution to better capture heavy-tails might be useful: Tails of Lipschitz Triangular Flows (https://arxiv.org/abs/1907.04481).